# Assessment of Water Mimosa (*Neptunia oleracea* Lour.) Morphological, Physiological, and Removal Efficiency for Phytoremediation of Arsenic-Polluted Water

**DOI:** 10.3390/plants9111500

**Published:** 2020-11-06

**Authors:** Narges Atabaki, Noor Azmi Shaharuddin, Siti Aqlima Ahmad, Rosimah Nulit, Rambod Abiri

**Affiliations:** 1Department of Biochemistry, Faculty of Biotechnology and Biomolecular Sciences, Universiti Putra Malaysia, UPM, Serdang 43400, Selangor, Malaysia; atabaki.narges@gmail.com (N.A.); aqlima@upm.edu.my (S.A.A.); 2Institute of Tropical Agriculture and Food Security, Universiti Putra Malaysia, UPM, Serdang 43400, Selangor, Malaysia; 3Department of Biology, Faculty of Science, Universiti Putra Malaysia, UPM, Serdang 43400, Selangor, Malaysia; rosimahn@upm.edu.my; 4Department of Forestry Science and Biodiversity, Faculty of Forestry and Environment, Universiti Putra Malaysia, UPM, Serdang 43400, Selangor, Malaysia; rambod.abiri@gmail.com

**Keywords:** phytoremediation, arsenic, *Neptunia oleracea*, removal efficiency, arsenic accumulation

## Abstract

Arsenic is considered to be a toxic and heavy metal that exists in drinking water and can lead to acute biotoxicity. Water mimosa (*Neptunia oleracea*) has been widely identified as a feasible phytoremediator to clean up aquatic systems. In the current study, the phytoremediation potential of water mimosa exposed to different concentrations of sodium heptahydrate arsenate (Na_2_HAsO_4_·7H_2_O) was tested. A number of plant physiological and growth responses such as height of frond, existence of green leaves, relative growth rate, relative water content, tolerance index, decrease in ratio of biomass and ratio of dry weight, chlorophyll content, photosynthesis rate, intercellular CO_2_ concentrations, stomatal conductance, air pressure deficit, transpiration rate, proline and lipid peroxidation, as well as arsenic accumulation and removal efficacy were analyzed. The micromorphological analysis results confirmed water mimosa’s tolerance of up to 30 ppm of arsenic treatment. The results obtained from the chlorophyll and gas exchange content also showed severe damage by arsenic at doses higher than 30 ppm. In addition, the highest arsenic accumulation and arsenic removal efficacy were observed at the range of 30–60 ppm. An analysis of proline and lipid peroxidation content confirmed water mimosa’s tolerance of up to 30 ppm of arsenic. The scanning electron microscopy (SEM) and X-ray spectroscopy (EDX) and analysis also confirmed the accumulation of arsenic as shown by the deformation of water mimosa tissues. The results showed that water mimosa is a reliable bioremediator for removing arsenic from aquatic systems.

## 1. Introduction

Environmental pollution is a phenomenon where air, water, and land become unsuitable or unsafe due to the existence of materials harmful towards living organisms [1,2]. Heavy metals including lead (Pb), thallium (Tl), arsenic (As), chromium (Cr), cadmium (Cd), and mercury (Hg) have been categorized as significant contaminants in the environment [3,4]. A toxic environment happens as a consequence of the release of heavy metal ions (even in small amounts) from mining, metallurgy, chemical manufacturing, and nuclear energy activities, thus, bringing extreme threats to the Earth’s crust [5]. 

Arsenic is an extremely toxic metalloid that is carcinogenic predominantly to humans’ liver, lungs, kidney, and bladder. It can also cause nerve damage and skin diseases [6]. This heavy metal is ubiquitous in the Earth’s crust with the potential effect of dietetic intake on human health in developed countries. It is also present through atypical groundwater exposure in developing countries and can contaminate the entire water source. These exposures are associated with human diseases, and toxicological concerning doses are comparable to typical dietary intake assessments [7]. The groundwater in the arsenic’s presence has affected the water supply to rural areas in over 70 countries. The estimations indicate that the number of people that have been exposed to arsenic is more than 150 million [8]. Inorganic arsenic accumulation in plants, livestock, and contaminated water can be transferred into the food chains [9]. The damage caused by this heavy metal on humans and the environment confirms that the elimination of arsenic from contaminated water is urgently essential. Removing arsenic from the environment through water and its downstream pathways is reportedly a complicated procedure [10]. Novel sustainable and innovative techniques can provide stable and efficient removal procedures of the metal from a water environment [11].

The rapid increase in human population, urbanization, industrial activities, deforestation, exploration, and exploitation of ecosystems has caused heavy metal and metalloid pollutions in Malaysia’s environment [12]. It has been detailed that the arsenic concentration is between 2.00 to 54.00 µg/L in rivers across Malaysia [13,14,15,16,17]. This is a concern as it exceeds the international environment guideline which is below 10 µg/L for drinking water samples [18]. To date, several treatment technologies have been announced for the arsenic removal from water bodies [19]. However, it is worth mentioning that the use of the most effective technology depends on plenty of factors such as environmental impact, operational cost and capital investment, the initial metal concentration, and plant reliability and flexibility [20]. 

Environmental contaminate removal from polluted water, sludge, soil, and sediments using plants is known as phytoremediation [21]. For the past two decades, phytoremediation has been developed as a green, non-invasive, and economic alternative to different conventional civil engineering-based strategies for the remediation of water, soil, and even residences contaminated with heavy metals [22,23]. Strategies employed under phytoremediation have included phytodegradation (employing plants or microorganisms to degrade contaminants) [24,25,26], phytoaccumulation (employing algae or plants to accumulate contaminants in their areal parts) [27], phytostabilization (employing plants to reduce the heavy metal mobility in soil) [28], phytofiltration (employing plants’ biomass and their associated rhizospheric microorganisms to refine contaminants) [27], phytovolatilization (employing plants to absorb contaminants and transpire them into the atmosphere in the volatile shape) [29], and rhizodegradation (employing plants to degrade contaminants using rhizosphere microbes’ mediation [30,31,32] (Figure 1).

Aquatic plants are natural candidates for treating contaminated soil and water by accumulating heavy metals in their tissues [34]. In the last five decades, the pantropical mimosoid legume genus Neptunia has awaken considerable interest mainly due to the aquatic habitat of some of its species [35]. The water mimosa (*Neptunia oleracea* Lour.) is an invasive and aggressive aquatic plant specific to Southeastern Asia, tropical Africa, and India [36]. Reportedly, *N. oleracea* has been broadly applied for the decontamination or reduction in contaminants in waters surrounding some Asian countries including Malaysia, Thailand, Indonesia, Philippines, and Vietnam [37]. *N. oleracea* is suitable for phytoremediation based on its short life cycle and growth ease, making it appropriate for planting in contaminated water areas [38]. A comparison with aboveground tissues has shown that heavy metals can accumulate in the root of *N. oleracea* [39]. The higher accumulation of heavy metals indicated removal efficiency of this plant through the rhizofiltration process [37]. This plant naturally grows in water bodies such as ponds and lakes, as well as rivers in Malaysia, therefore, providing some advantages in their planting and application such as high level of treatment, being evergreen, has high biomass production, has good adaptation to the tropical climate, is inexpensive, and requires simple maintenance [40]. 

Given the above, the main objective of the current study is to elucidate the influence of different arsenic contaminations on the morphological, physiological, as well as histological characteristics of *N. oleracea* lour. In addition, the present investigation aims to provide insight regarding removal efficiency of water mimosa for phytoremediation of arsenic-polluted water.

## 2. Results and Discussion 

### 2.1. Macromorphological Observation of Water Mimosa under Different Arsenic Concentrations

In the current study, a hydroponic experiment was carried out to evaluate the impact of sodium heptahydrate arsenate (Na_2_HAsO_4_·7H_2_O) on the growth performance and development of *N. oleracea* (water mimosa). Furthermore, the phytoremediation potential was tested by measuring the water mimosa’s ability to accumulate arsenic. In this study, nondestructive morphological observations were measured on the first and 14th days. Quantitative and qualitative observations of the plants on the first and 14th days showed that the growth and reactions of the water mimosas varied in response to the different arsenic treatment concentrations (Figure 2). The addition of arsenic negatively influenced the morphological appearance of treated water mimosas. Increased arsenic concentrations over time caused the number and growth ratio of leaves and roots, as well as root and shoot diameters decreased. Morphological observations showed that after 14 days, water mimosas were resistant to low levels of arsenic concentrations (less than 60 ppm) (Figure 2i,m,n,o). Nonetheless, increasing the arsenic concentrations to 60 ppm had severe effects which caused serious damage followed by the death of the plants (Figure 2q–t). Increasing the arsenic levels to more than 70 ppm led to significant changes in the plants’ characteristics. At higher arsenic concentrations, the morphological analysis showed severe symptoms of damage, and ultimately, the death of the plants (Figure 2r–t).

Measurement of the plants’ morphological traits was the initial step taken to observe the phytoremediation potential of water mimosas against heavy metals. In Cd-treated *Bromus kopetdaghensis*, decreasing trends were reported for a majority of morphological characteristics such as root and shoot height, and dry weight [41]. Similarly, reductions in the morphological traits of *Prosopis laevigata* were reported under heavy metal treatment. It has been documented that plants with phytoremediation properties absorbed heavy metals, and this absorbance could cause severe damage at higher levels of toxicities. The colors of leaves and plants, number of leaves, and structure of roots and shoots are part of the morphological features that can be observed to degrade at the early stages of toxicity [42]. Since water mimosas are fast-growing aquatic plants, they can absorb nutrients and heavy metal rapidly. The absorbed nutrients/heavy metal can be transferred easily to the areal parts and leaves and present their impact on the morphology of the plant [37]. In this study, the color of leaves had turned yellow and, after seven days, wilting had occurred. Increasing the treatment time up to 14 days caused the plants’ color to change to brownish, followed by the plants’ death at the final stage. The roots of water mimosas placed under control conditions were thinner than the roots of plants treated with arsenic (woody roots) (Figure 3). After increasing the arsenic level, the leaves’ color changed from greenish to yellowish and, ultimately, died. The changes in leaves’ morphology started at the early stages of treatment, but significant changes were observed on Day 14. Leaves had dropped for water mimosas treated with arsenic concentrations of 50 ppm and above, at Day 7. At Day 14, the arsenic’s toxicity was observed clearly through the leaves’ survival rate and color at higher concentrations. At the end of the experiment, the color of roots changed from pinkish to brownish. The leaves of the control plants remained green and normal in shape. The stems of plants under control were straight; however, the stems of plants treated with arsenic were near the surface of the tanks. It has been reported that the leaves of water mimosas make rapid movements in response to touch [43]. The leaves’ movement showed different reactions in response to light and strong touch. Water flux across the tonoplast/plasma membrane, and the parallel leaves reaction are relevant to the water channel aquaporin as a particular membrane protein [44]. In this experiment, the authors observed that the leaves of water mimosas treated with arsenic showed the same reactions when they were touched. For lower levels of arsenic treatment at shorter treatment periods, the leaves of the water mimosas were folded at points (*pulvinules*) along the rib (*rachis*). However, when the levels of arsenic were increased for longer durations, the branches and leaves dropped together at the *pulvinus* zone where the main branch (*petiole*) joins the stem [44]. 

### 2.2. Physiological Changes of Water Mimosa under Arsenic Treatment

#### 2.2.1. Arsenic’s Impact on Micromorphological Traits of Water Mimosa 

ANOVA and Duncan’s multiple comparison tests of the arsenic impact on the water mimosas were significant (*p* ≤ 0.01) in terms of decreasing ratio of biomass (DRB) and decreasing ratio of dry weight (DRD). Nonsignificant differences were observed between the replicates (Table 1). 

The most considerable biomass belonged to the control plants where the least decreasing ratio of biomass (DRB) recorded was 0.69% (Figure 4a). However, the comparison with the control plants showed that the lowest DRB observed among plants treated with 5 ppm arsenic was a 6.7% reduction after 14 days. Additionally, the highest reduction in the DRB belonged to plants treated with 100 ppm arsenic at 93.7%. The DRB significantly increased up to 70 ppm arsenic concentration followed by a decreasing trend at 90 ppm and another increasing rate at 100 ppm. According to Duncan’s multiple comparison test, plants treated with 30 and 50 ppm arsenic treatments were in the same group (non-significant differences), while those treated using 70, 80, and 100 ppm concentrations of arsenic were in the same group, and non-significant differences were observed (Figure 4a). The highest dry weight belonged to the control plants in which the least decreasing ratio of dry weight (DRD) was observed in water mimosas at 0.7% (Figure 4b). Although the highest DRD was observed for plants treated in 100 ppm (93.75%) arsenic followed by 80 mL/L arsenic (92.93%), non-significant differences were seen for plants treated using 70, 80, 90, and 100 ppm arsenic concentrations (Figure 4b). Comparing all treatments, the lowest DRD was reported for plants under 5 ppm arsenic concentration at 16.61% (Figure 4b). The DRD significantly increased for plants treated up to 70 ppm concentrations of arsenic, followed by a slight increase in the rest of the concentrations (Figure 4b). 

The effects of arsenic on the height of frond percentage, green leaves percentage, relative growth rate (RGR) percentage, relative water content (RWC), and tolerance index (Ti) of water mimosas at different concentrations are reported in Table 2. The highest height of frond was observed in control plants (5%) and the lowest height of frond was reported for plants under 100 ppm arsenic treatment after 14 days (Table 2). However, a non-significant difference was observed between the control plants and plants treated with 5 ppm arsenic (Table 2). The percentage of green leaves also decreased after 14 days with increasing arsenic concentrations (Table 2). The RGR in water mimosas presented decreased ratios with an increase in arsenic concentrations. The RGR for water mimosas under the control condition (no arsenic) was 0.004 g/g·day, a two-fold decrease (0.002 g/g·day) from the plants treated with 5 ppm arsenic. A decreasing trend was also reported in water mimosas under 10 ppm arsenic treatment. However, no PGR was observed in plants under the highest levels of arsenic treatment (Table 2). The highest RWC was observed in water mimosas treated with 100 ppm arsenic and the lowest RWC was reported in plants under control (free of arsenic). In addition, the highest and lowest tolerance indices were presented in the control plants and water mimosas under 100 ppm arsenic treatment, respectively (Table 2).

Each of the arsenic concentrations applied to the water mimosas had significant detrimental effects on the plants’ growth and development. These results are in parallel with reports on other high-tolerance species such as *Salix purpurea* (Purple willow) [45], *Pteris vittata* (Chinese brake) [46], *Oryza sativa* (rice) [47], and *Typha latifolia* [48]. It has been shown that severe concentration of arsenic dosages inhibited the normal life cycle of plants and in the detrimental phase, could kill plants. Changes in the dry and fresh weights of plants under arsenic treatment revealed that arsenic directly affected photosynthesis, the life cycle of cells, and plants’ receptors [49,50]. Generally, plants can tolerate heavy metals like arsenic using developed mechanisms such as transporting heavy metals through internal tolerance mechanisms or limiting the absorption by cascades of changes from a plant’s receptors and maintaining its cell structure [51]. Although increased concentrations of heavy metals such as arsenic decreased plants’ tolerance, it has been shown that plants with tolerance indices of more than 60% were assumed to be good tolerant bioreactors [52]. The results demonstrated good tolerance to arsenic by water mimosa plants exposed to 30 ppm after 14 days. However, the Ti value of the water mimosas showed that the plant also had good tolerance to arsenic after exposed to 60 ppm over 7 days (data are not presented). The decreasing trend of RGR was reported due to an increase in biomass over time [53]. The combined micro-morphological data analyses of Ti, RWC, RGR, green leaves, and the height of frond indicate that water mimosas display good capacity for growth, metal bioconcentration, and tolerance up to 14 days under 30 ppm of arsenic exposure. 

#### 2.2.2. Impact of Arsenic on Physiological Traits of Water Mimosa 

The ANOVA and Duncan’s multiple comparison tests on the water mimosas were significant (*p* ≤ 0.01) in terms of photosynthesis rate, stomatal conductance, transpiration rate, and chlorophyll content. Non-significant differences were observed between the replicates (Table 3). The ANOVA analysis of water mimosa also showed significant differences at the 5% level after seven days, and significant differences at the 1% level after 14 days of intercellular CO_2_ concentration (Table 3). The ANOVA analysis also showed non-significant differences after seven days and significant differences (*p* ≤ 0.05) after 14 days of air pressure deficit (Table 3). 

The results of the photosynthetic activities test showed that the highest net photosynthesis rate was observed for the controlled water mimosas at 11.69 µmol CO_2_ m^−2^·s^−1^ (Figure 5a). A comparison of all treatments showed that the highest photosynthesis rate of the water mimosa plants was obtained from those treated with 5 ppm arsenic at 0.99 µmol CO_2_ m^−2^·s^−1^, while the lowest total photosynthesis rate was recorded for those treated in 100 ppm arsenic at 0.12 µmol CO_2_ m^−2^·s^−1^. A decreasing trend of photosynthesis rate was also observed in *Bambusa vulgaris*. At the detrimental phases, the heavy metal destroyed the leaves completely and led to the plants’ death [54]. In other reports, arsenic has been shown to cause oxidative stress, a decrease in photosynthesis rate, and inhibit growth parameters in Neem (*Azadirachta indica*) and Tulsi *(Ocimum sanctum*). These changes occurred due to a decrease in leaf sizes, alteration of stomatal pores, and deformation of leaves [55]. Although the photosynthetic rates of some rice cultivars increased, intercellular CO_2_ concentrations, conductance to H_2_O, and transpiration rate decreased under arsenic stress [56]. Additionally, the highest stomatal conductance rates were obtained for control plants at 0.17 mol H_2_O m^−2^·s^−1^ followed by those treated with 5 ppm of arsenic at 0.084 mol H_2_O m^−2^·s^−1^, whereas the lowest total stomatal conductance rate was recorded in plants treated with 100 ppm arsenic at 0.007 mol H_2_O m^−2^·s^−1^ after 14 days (Figure 5b). Stomatal conductance under high concentrations of arsenic was slightly reduced on Day 14 (Figure 5b). 

Therefore, arsenic has been proven to directly or indirectly affect plants’ photosynthetic functions. This suggests that under heavy metal stress, carbon assimilation is reduced, a situation which can directly inhibit plant growth. The arsenic treatment caused a considerable decreasing trend of stomatal conductance which could be the consequence of stomatal closure or nonstomatal inhibition of photosynthesis [54]. This reduced rate of stomatal conductance could also have been caused by leaf damage and subtended stomata in treated plants. These results were similar to the results of an assessment on *B. vulgaris* after exposure to heavy metals [54]. Low stomatal conductance, root absorption rate, water content, compatible solute of arsenic accumulation, osmotic potential, and leaf conducting tissues were reported as general features of plants under heavy metal stress [57].

The highest intercellular CO_2_ concentration was reported for plants treated with 90 ppm arsenic concentration at 527.82 μmol CO_2_ mol^−1^ after 7 days (Figure 5c). However, the lowest intercellular CO_2_ concentration for water mimosas was recorded for plants treated with 5 ppm arsenic at 201.08 μmol CO_2_ mol^−1^ after 7 days. The highest intercellular CO_2_ concentration was observed for plants under 90 ppm arsenic concentration at 367.96 μmol CO_2_ mol^−1^ after 14 days (Figure 5c). However, the lowest intercellular CO_2_ concentration for water mimosas was recorded for those under 60 ppm arsenic concentration at 93.47 μmol CO_2_ mol^−1^, followed by plants under 50 ppm arsenic concentration at 94.60 μmol CO_2_ mol^−1^ after 14 days. The intercellular CO_2_ concentration decreased slightly on Day 14, especially for plants under higher concentrations of arsenic treatment. The results of an investigation on *O. sativa* showed that intercellular CO_2_ concentrations, conductance to H_2_O, and transpiration rate had significantly decreased after treatment [56]. These findings were in the same trend as the results on water hyacinth after 14 days of arsenic treatment [56]. In a study on other heavy metals’ effects on intercellular CO_2_ concentrations, lettuce plants (*Lactuca sativa*) exposed to low doses of Pb(NO_3_)_2_ for 28 days showed negative effects in terms of CO_2_ assimilation. However, the transpiration rate, intercellular CO_2_ concentration, and stomatal conductance were not affected, and the lettuce plants did not display clear growth impairment or even morphological changes [58]. These results were similar to the response obtained from water mimosas treated with different arsenic concentrations in terms of intercellular CO_2_ concentrations after seven days of treatment. 

The highest transpiration rate was obtained in control samples at 3.31 mmol H_2_O m^−2^·s^−1^ followed by those under 50 ppm arsenic treatment at 1.13 mmol H_2_O m^−2^·s^−1^. Contrariwise, the lowest transpiration rate was recorded in plants treated with 90 ppm arsenic at 0.14 mmol H_2_O m^−2^·s^−1^, followed by those treated with 100 ppm arsenic at 0.15 mmol H_2_O m^−2^·s^−1^ (Figure 5d). Arsenic accumulation also caused significant poor growth due to enhancements to the peroxidation lipid and content hydrogen peroxide which led to a significant reduction in the transpiration rate, photosynthesis rate, and stomatal conductance of *Ricinus communis* genotypes [59,60]. Inhibition of the transpiration rate was caused by stomata closure [61,62], xylem embolism [63], and leaf damage [64]. In the current research, the results indicated that plants treated with different concentrations of arsenic suffered a significant decline in transpiration rate as compared with the controls. The same results in gas exchange attributes were observed in a study that focused on the effect of arsenic toxicity on the photosynthesis growth of rice seedlings, and recorded that higher durations of arsenic treatment caused a greater decline in transpiration rates [55].

The highest air pressure deficit in water mimosas was obtained from plants treated with 5 ppm concentration of arsenic at 2.29 kPa after seven days of treatment (Figure 5e). Contrariwise, the lowest air pressure deficit was recorded for the control plants followed by those treated with 50 ppm arsenic after seven days of treatment at 1.72 and 1.97 kPa, respectively. On the one hand, the highest air pressure deficit in water mimosas was obtained from plants under 30 ppm arsenic concentration followed by those under 5 ppm arsenic concentration after seven days of treatment at 2.06 and 2.05 kPa, respectively. On the other hand, the lowest air pressure deficit was recorded for plants treated with 70 ppm arsenic followed by those under 80 ppm arsenic treatment after seven days of treatment at 1.68 and 1.69 kPa, respectively. The results indicated that water mimosas reacted most sensitively to metal pollution through significant reductions in gas exchange. Similar responses in terms of photosynthesis rate, stomatal conductance, and intercellular CO_2_ concentration, in this investigation, suggest that the different photosynthetic responses of both plant treatments to excess arsenic might be stronger due to the low pigment content and stomatal conductance stress caused by high arsenic toxicity. Arsenic exposure as an abiotic stress could also cause direct or indirect alternation and damage to plant cells over the construction of reactive oxygen species (ROS) [65]. During growth, the vapor pressure deficit has a slight impact on the transpiration efficiency of different genotypes at whole-plant and leaf levels [66]. The results of this study revealed that high concentrations of arsenic promote da higher negative impact on the vapor pressure deficit. These results were similar to a study on *Bambusa vulgaris* which indicated that *B. vulgaris* was affected by high heavy metal concentrations as exhibited by the lowest Vpdl recorded at the end of the arsenic treatment period [54]. A study on *Zea mays*’ vapor pressure deficit recorded a 75% reduction as compared with respective stress controls [67].

The highest chlorophyll content for water mimosas was reported for plants under 10 ppm arsenic concentration at 34.26 mg/cm^3^ after seven days (Figure 5f). However, the lowest chlorophyll content was recorded for plants under 100 ppm concentration of arsenic after seven days at 10.93 mg/cm^3^. In water mimosas, the highest chlorophyll content was observed in plants under 10 ppm arsenic concentration at 33.56 mg/cm^3^ after 14 days. However, the lowest chlorophyll content for water mimosas was recorded for those treated with 100 ppm arsenic at 6.03 mg/cm^3^ followed by water mimosas under 50 ppm arsenic concentration after 14 days at 94.60 mg/cm^3^. Chlorophyll content as a biochemical parameter for water fern (*Salvinia natans*) was estimated along with exposure time to observe the variations in *Salvinia* biochemical constituents. The study showed that at the earliest contact hours, no significant negative impact on chlorophyll content was observed for plants under 2.0 ppm first arsenic concentrations. However, a significant toxicity effect was obtained in the form of a 60% chlorophyll content loss after five days, even for initial arsenic concentrations. These results demonstrate the inactivation of the photosystem electron transport and a chloroplast membrane disorder due to arsenic treatments [68]. In *B. vulgaris*, the lowest chlorophyll content was observed at the highest heavy arsenic concentration (300 ppm) at the end of the treatment period [54].

Analysis of the water mimosas under different arsenic treatments on Day 14 showed various proline and MDA (lipid peroxidation) contents. The highest level of MDA was observed in 50 ppm at 35.47 ± 1.8 and 37.23 ± 0.80 (µmol/g FW) for root and leaves, respectively (Table 4). Additionally, the level of MDA in the leaves was higher than in the roots under all arsenic treatments (Table 4). Subsequently, a higher level of proline was observed in the roots (35 ± 0.4 µmol/g FW) and leaves (43 ± 1.3 µmol/g FW) of the sample treated with 30 ppm arsenic (Table 4). In the treated root samples, the lowest levels of proline were observed for plants treated with 90 ppm (19 ± 0.4 µmol/g FW) and 100 ppm (19 ± 0.5 µmol/g FW) arsenic (Table 4). 

The results strongly suggest that arsenic toxicity affects some of the vital enzymes needed for the antioxidant defence mechanism of water mimosas. It has been shown that MDA content was one of the most important mechanisms which could lead to plants’ resistance against oxidative stresses, and therefore adaptation and increased survival rate under adverse conditions [69]. MDA is a reactive aldehyde which is produced by lipid peroxidation and is boosted under adverse conditions. Reportedly, MDA has been focused on as an oxidative stress indicator [70]. The MDA content could increase under heavy metal stress due to the concentration-dependent free radical production. The better the oxidative damage protection is, the more quickly the antioxidative system could be to upregulate. Peroxidizing activity can lead to the MDA elimination. An increase in antioxidative enzyme activity, such as peroxidizing activity, leads to MDA elimination, and subsequently, a reduction in H_2_O_2_ amount and the membrane damage [69]. These results showed that water mimosa was able to tolerate arsenic up to 30–50 ppm (Table 4), in spite of the fact that lipid peroxidation was enhanced by this metalloid and, as a consequence, the cell membrane stability was affected [71,72]. The arsenic-dependent decrease in MDA accumulation in the range of 60–100 ppm could be attributed to the reduction in the survival rate of plants. Similarly, accumulation of proline is another index in a plant’s defence mechanism against stresses. Proline is an osmo-compatible solute which raises the ability of water mimosas and other phytoremediation plants to endure toxicity from arsenic and other heavy metals [73]. In the current study, proline accumulation in the roots and leaves first increased to a peak value of 35 ± 0.4 (µmol/g FW) and 43 ± 1.3 (µmol/g FW), respectively, and then decreased. The results indicated that water mimosas could tolerate up to 30 ppm of arsenic; after that, arsenic would damage the plant’s mechanisms. Reportedly, the relationship between proline accumulation and arsenic level could be of a substrate-product nature, due to the direct or indirect impact of heavy metal stresses on proline biosynthesis [74]. This might be due to the relationship between proline and arsenic diverting from glutamate and practical indirect activities identifying with sub-products of arsenic catabolism such as gamma-aminobutyric acid (GABA) and H_2_O_2_ [75]. Interestingly, the stress-induced cellular acidification was reduced due to the proline’s accumulation under stress. Proline might act as a singlet oxygen scavenger and hydroxyl radical, a carbon and nitrogen source required in stress recovery and a component of stress signal transduction mechanisms, thus contributing to the development of heavy metal tolerance [76].

### 2.3. Phytoremediation Attributes of Water Mimosa under Arsenic Treatment

#### 2.3.1. Arsenic Accumulation of Water Mimosa 

The ANOVA and Duncan’s multiple comparison tests on the water mimosas were significant (*p* ≤ 0.01) in terms of ICP of the water mimosa plant removal efficiency after seven and 14 days. Non-significant differences were observed between the replicates (Table 5).

As shown in Figure 5, the water mimosas accumulated arsenic in their roots up to 2.8 mg·Kg^−1^, even at the lowest concentration of arsenic (5 ppm) during the 14 days of experimentation. The water mimosas presented severe necrotic symptoms (Figure 2 and Figure 3) at the highest concentrated condition (100 ppm) after 14 days of treatment. However, the plant was able to accumulate 16.36 mg·Kg^−1^ arsenic in its roots (0.8 mg total arsenic) (Figure 6). Plants exposed to 70 mg·L^−1^ arsenic accumulated a concentration of 17 mg·Kg^−1^ in their roots (0.97 mg total arsenic) and the highest accumulation was recorded at 30 ppm with 28.192 mg·Kg^−1^ (0.3 mg total arsenic). 

Although arsenic has been recorded at the lowest arsenic treatment, small amounts of arsenic potentially have been translocated from the growth solution to the leaves [45]. A comparison of the amount of arsenic in plants and water showed that almost all arsenic accumulation was retained in the roots at 70 ppm and only a small amount was translocated to the shoots. These findings are similar to previous studies on some hyper-accumulator plants with the capability of translocating arsenic from roots to above-ground [77]. However, this plant also showed general plant dysfunction at high concentrations. The same results were achieved in a study on purple willow (*Salix purpurea*) under 30 and 100 ppm arsenic concentrations [45]. Interestingly, water hyacinths treated with 70 and 100 ppm arsenic showed the same absorption of foliar-applied arsenic with the arsenic hyper-accumulating fern (*Pteris vittata* L.) [78]. Another examination on water hyacinth found a maximum uptake of arsenic at 0.0309 mg/g in its dry plant tissue [49]. In microorganisms, detoxification operons have been demonstrated as the most common arsenic resistance form against heavy metals [79]. In another study, two genes (*asoA* and *asoB*) were encoded as the subunit of oxidation of arsenite in *Alcaligenes faecalis* which was involved in metabolism processes and arsenic resistance [80]. It has been shown that, in *Sulfurospirillum barnesi*, the resistance mechanism of a single operon was encoded in the cell membrane where the reduction in arsenate occurs [81]. In previous reports, a majority of up-taken arsenic accumulated in the roots, thus providing reason for the authors to focus on the roots of each water mimosa plant [45,77].

#### 2.3.2. Arsenic Effect on Removal Efficiency of Water Mimosa 

In water mimosas, the highest removal efficiency was observed for plants under 60 ppm arsenic concentration at 13.73% after seven days of arsenic treatment (Figure 7). However, the lowest removal efficiency for water mimosas was recorded for those treated with 5 ppm arsenic after seven days at 6.7% (Figure 7). The highest removal efficiency was reported for water mimosas under 60 ppm arsenic concentration at 17.43% followed by those treated with 30 ppm arsenic at 17.24% after 14 days of treatment (Figure 7). Nevertheless, the lowest removal efficiency for water mimosas was recorded for plants under 5 ppm arsenic concentration at 8.5% after 14 days of arsenic treatment (Figure 7). 

In regard to the arsenic removal percentage at different concentrations, significant differences were recorded among the treatments. These findings are in line with previous studies, for example, *Echinodorus cordifolius* was found to have the highest arsenic removal efficiency, followed by some other aquatic plants, for instance *Cyperus alternifolius, Acrostichum aureum* and *Colocasia esculenta.*, respectively [82]. The removal of arsenic was up to 38.8% for water hyacinth; the result paralleled another experiment carried out by Ingole and Bhole [49] who recorded a 32% removal efficiency for water hyacinth when arsenic was present at an initial concentration 5 ppm. In the case of water mimosas’ heavy metal uptake, according to previous research, the highest removal levels were for cadmium and lead as compared with other heavy metals [83]. This suggests that the removal rate of arsenic increases as the concentration increases, due to higher arsenic absorption by the roots into the plants, as has been proven by Darajeh et al. [84] using plant root length. Concentrations in plants and high removal efficiency may happen due to a plant’s high biomass during growth, in addition to the metallophilic root system’s proliferation associated with the possibility of arsenic cross-contamination from external sources that may lead to higher arsenic uptake [85].

#### 2.3.3. Impact of Arsenic on Histology of Water Mimosa 

X-ray spectroscopy (EDX) (LEO 1455 VPSEM, London, UK) and scanning electron microscopy (SEM) observations were done on the roots of the control and 30 ppm treated water mimosas after 14 days of treatment (Figure 8). The roots of the control plants were thinner than the roots of plants treated with arsenic (woody roots). The hairy roots of the control samples were thinner and young (Figure 8a), but the hairy roots of treated samples were woody and thick (Figure 8f). The epidermis of both control and treated samples was multi-layered. There were several parenchyma tissues with a rectangular structure in the control samples (Figure 8b–d). However, the parenchyma tissues had an irregular structure in the treated samples (Figure 8f–i). In the control samples, the cortex (outer layer of plant’s root) generated circular intercellular nodules on its parenchyma cells. As compared with the control plants, the cortex of the treated samples were composed of irregular intercellular nodules on its parenchyma cells (Figure 8). The SEM analysis also proved the availability of arsenic in treated samples as compared with the controls (Figure 8j). The scanning electron microscopic analysis of *Mimosa calodendron* under arsenic treatment showed dissimilar changes in cell size. Arsenic can make visible changes to cell volume. This significant increase in the volume of the cell could be interpreted as a possible defence mechanism of the cell to avoid damage from arsenic toxicities through increasing arsenic acclamation [86].

## 3. Materials and Methods

### 3.1. Plant Materials and Culture Conditions

Local naturally grown *N. oleracea* aquatic plants with similar weight and size were collected from Universiti Putra Malaysia’s pond (2°59′23.8″ N, 101°42′46.5″ E), and the plant species was confirmed by the Biodiversity Unit (UBD) at the Institute of Bioscience, Universiti Putra Malaysia. The collected plants were acclimatized under hydroponic conditions in tanks (12 × 25 × 10 cm = 3 L) containing 0.20× of Hoagland solution with an aeration system. 

### 3.2. Arsenic Stress Treatment

After one month of acclimatization, each plant was exposed to different levels of sodium heptahydrate arsenate (Na_2_HAsO_4_·7H_2_O) concentrations as follows: 0 (control), 5 (67), 10 (135), 30 (400), 50 (667), 60 (801) 70 (933), 80 (1067), 90 (1200), and 100 (1335) ppm (µM), and the treated plants were kept, for two weeks, at 18–25 °C with an 18 h light/6 h dark photoperiod under a light intensity of 500 µmol·m^−2^. 

### 3.3. Arsenic Stock Solution Preparation 

Different concentrations of sodium heptahydrate arsenate (Na_2_HAsO_4_·7H_2_O) were measured based on the following equation: Concentration of Na_2_HAsO_4_·7H_2_O% = (ppm range/molecular weight percentage of As) × 100(1)

For the preparation of the 1000 ppm stock solution of arsenic, 4.16 g of Na_2_HAsO_4_·7H_2_O was dissolved in 1000 mL of distilled water [45]. The application of treatment was randomized, and each tank’s volume was maintained over 2 weeks to 3 L only by adding distilled water. The total amount of arsenic applied was 0, 15, 30, 90, 150, 180, 210, 240, 270, and 300 mg. 

### 3.4. Morphological Attribute Evaluation of N. oleracea under Arsenic Treatment

Plant growth and physiological parameters were measured on the 1st and 14th day of plant treatment. First, the area/length, fresh weight and height, and growth ratio of the water mimosas’ different parts were measured using a ruler, weight scale, and through visual observations. The root and shoot diameters were recorded using a digital Vernier calliper (Mitutoyo UK Ltd., Hampshire, UK).

### 3.5. Relative Growth Rates

The relative growth rates (RGR) of the water mimosas were measured as per the formula written in Equation (2) [87].
RGR = (lnW_1_ − lnW_0_)/(t_1_ − t_0_)(2)
where W_0_ is the initial and W_1_ is the final weights, while t_1_ is the beginning and t_0_ is the end of the treatment duration. 

### 3.6. Decreasing Ratio of Biomass (DRB) and Decreasing Ratio of Dry Weight (DRD)

The decreasing ratio of biomass (DRB) and decreasing ratio of dry weight (DRD) were calculated based on the modified formula by Abiri et al. [88] as:
DRD = (Drought weight of plants on day 14 × 100)/(drought weight of plants on day 1) − 100(3)

### 3.7. Relative Water Content

The relative water content (RWC) was determined, according to Chen et al.’s [89] formula, as follows:
RWC (%) = [(FW − DW)/FW] × 100(4)
where FW is the wet plant biomass which was measured immediately, and DW is the dry weight biomass of samples in an oven at 65 °C after 48 h [40].

### 3.8. Tolerance Index

The tolerance index (Ti) was measured as follows: Ti = (dry weight treated plant (g)/dry weight control plant (g)) × 100(5)
where water mimosas with Ti ≥ 60% were reflected as highly tolerant [40].

### 3.9. Physiological Features Assessment of N. oleracea under Arsenic Treatment

#### 3.9.1. Chlorophyll Contents 

The chlorophyll contents of the leaves were collected in the morning at the same time weekly using a Konica Minolta SPAD 502Plus Chlorophyll Meter (Konica Minolta China Investment Ltd., Shanghai, China) which is a lightweight handheld meter that can be used without causing any damage to plants. 

#### 3.9.2. Gas Exchange Attribute

The fully expanded leaves of each plant were selected for this experiment. Gas exchange parameters including net photosynthesis rate (Anet), transpiration rate (E), stomatal conductance (Gs), intercellular CO2 (Ci), and leaf to air vapor pressure deficit (VpdL) were measured using a LiCor 6400 Portable Photosynthesis System (LiCor, Inc., Lincoln, NE, USA). This open-type photosynthesis system was equipped with a standard 3 × 2 cm broadleaf cuvette. Calibration of the flow meter and CO_2_ zero values was made before the gas exchange was measured. Then, the CO_2_ concentration was set at 360 mol·m^−2^·s^−1^ to avoid any effect from fluctuating environmental conditions. Cuvette irradiance, temperature, and relative humidity were set at 650 µmol photons m^−1^·s^−1^ (saturating irradiance), 25 °C, and 40%, respectively. 

#### 3.9.3. Proline Contents

According to Bates et al. [90], the proline content of plants under control or arsenic stress was measured using the ninhydrin acid reagent. Briefly, the sample collection for proline content was done on Day 14 and in this regard, 500 mg of leaf tissues were detached and homogenized in a cold mortar and pestle by adding 3% *w*/*v* sulfosalicylic acid (5 mL). Next, the homogenized phases were centrifuged at 10,000× *g* for 15 min. In the next step, 2 mL of the acid ninhydrin were added to the 2 mL of supernatant, and 2 mL glacial acetic acid. The preparation of acid-ninhydrin was done by warming agitation with 30 mL glacial acetic acid, 1.25 g ninhydrin, and 20 mL 6 M phosphoric acid, until dissolved. Subsequently, the mixture was cooled at room temperature (4 °C) for 24 h by incubating the mixture at 100 °C for 30 min, the red brick color developed. Finally, 4 mL of toluene was transferred into the tubes which were vortexed for 30 s. The top layer (chromophore containing toluene) was isolated and the absorbance was recorded at 520 nm in the spectrophotometer against blank toluene. The proline concentration was measured using the standard curve of L-proline [90,91]. 

#### 3.9.4. Lipid Peroxidation Contents 

Lipid peroxidation (malondialdehyde (MDA)) was calculated, based on Davenport et al. [90]. About 200 mg of root and leaves were collected and homogenized in 2 mL of 5% (*w*/*v*) trichloroacetic acid and, subsequently, centrifuged at 10,000 rpm for 10 min at 4 °C. In the next step, 2 mL of 0.67% (*w*/*v*) thiobarbituric acid was added into 2 mL of supernatant and the incubating of the mixture was done in a boiling water bath for 30 min and, subsequently, centrifuged after cooling. The supernatant absorption was done at 450, 532, and 600 nm. The MDA content was calculated as described below:MDA (µmol·g^−1^) = [6.45 × (A532 − A600) − (0.56 × A450)] × Vt/W(6)
where Vt = 0.0021 and W = 0.2.

### 3.10. Arsenic Content Analysis Using Inductively Coupled Plasma Optical Emission Spectrometry (ICP-OES)

#### 3.10.1. Plant Sample Preparation and Acid Digestion Method 

The roots of control and treated plants were detached and weighed after 14 days of treatment. Whole samples were dried at 70 °C for two days in oven. The smaller pieces of dried tissues were sieved (2 mm size). All powdered samples were cooled down, and then accurately weighed at 0.5 g, 5 mL of 10% HCl was added to their container. To obtain a clean solution, the containers with the acid solution, was kept on a hot plate and digested. The dissolving of the final residue was done in 10 mL of 20% HNO_3_ solution and boiled at 100 °C for one hour. Then, the solution was cooled and transferred quantitatively to a 50 mL volumetric flask by adding distilled water. Whatman’s 42 filter paper was used for the samples’ final filtration before the determination of the samples’ metal concentrations with an ICP-OES Optima 7300 Perkin Elmer [92]. 

#### 3.10.2. Analysis of Arsenic Uptake from Water

Water samples were taken from both control (0 mg·L^−1^ of arsenic) and treated tanks at the 0th, 7th, and 14th days and kept at 4 °C to evaluate their arsenic content. The level of water inside each tank was kept constant by adding distilled water. Samplings were done at the same time and the arsenic contents of samples were analyzed using the ICP-OES. 

The removal efficiency of arsenic was calculated, according to Darajeh et al. [84] equation as follows:% Removal efficiency = (C_ini_ − C_fin_)/C_ini_ × 100(7)
where C_ini_ is the initial concentration of synthetic mixture and C_fin_ is the final concentration of the synthetic mixture.

### 3.11. Detection of Arsenic in N. oleracea Samples Using Electron Microscopy Analysis 

#### 3.11.1. Sample Preparation for Electron Microscopy

Roots (1 cm^2^ slices) were cut, put into separate vials, and fixed in 4% glutaraldehyde (fixative) for 2 days at 4 °C. Then, the samples were washed with 0.1 M sodium cacodylate buffer (NaO_2_As(CH_3_)2·3H_2_O) for 3 changes of 30 min. Post-fixation was done using 1% osmium tetroxide (OsO_4_) for 2 h, at 4 °C. Rewashing was performed using 0.1 M sodium cacodylate buffer for 3 changes of 30 min. Dehydration was implemented using a series of acetone concentrations at different times. 

#### 3.11.2. Localization of Sodium Hepta Hydrate Arsenate in Treated Samples

Images and quantitative analysis of plant samples under sodium heptahydrate arsenate (Na_2_HAsO_4_·7H_2_O) were obtained using energy-dispersive X-ray spectroscopy (EDX) (Leo 1455 VP-SEM, New England, USA) and scanning electron microscopy (SEM). To assess the amount of arsenic using EDX, energy to wavelengths were counted as follows:Wavelength (A) = 12.3983/Energy (keV)(8)

Three independent replicates of treated and control plants were utilized to show the consequence of accumulation of sodium heptahydrate arsenate in the roots of individual plant samples. To measure the arsenic contents of root samples, three different spectrums of roots were randomly measured for each image. The magnification and accelerating voltage of images were 3509 and 2000 kV, respectively [88].

### 3.12. Statistical Analysis 

For morphological and physiological studies, a randomized complete block design (RCBD) was used for a total of 30 tanks distributed in 3 experimental blocks (10 arsenic concentrations × 3 replications). For the analysis of the data, the SAS software version 9.4 was applied. The level of significance was assessed from the analysis of variance (ANOVA). Duncan’s multiple ranges were used to compare the mean values, and interpretations were made accordingly.

## 4. Conclusions 

In summary, this investigation provided insight into the effect of arsenic stress on water mimosas and the plants’ response. The results of the micromorphological analysis along with the qualitative observations proved water mimosas’ tolerance up to 30 ppm arsenic treatment. Increasing the arsenic treatment above 30 ppm caused all of the plants’ micromorphology to be damaged. The physiological traits analysis showed the toxicity effects of arsenic even at the initial stages of the experiment, which caused damage to the chlorophyll content, photosynthesis rate, stomatal conductance, intercellular CO_2_ concentrations, transpiration rate, and air pressure deficit. Observations on the proline content showed water mimosas’ adequate tolerance under 30 ppm of arsenic treatment. The inductively coupled plasma optical emission spectrometry (ICP-OES) showed the accumulation of arsenic in water mimosas in the range of 30–60 ppm. However, increases in the arsenic’s toxicity caused the accumulation rate to decrease. The link between this step with the previous physiological micromorphological traits showed that at higher levels of arsenic toxicity, plants were highly likely to die, therefore, the decreasing trend of arsenic accumulation might be due to this decreasing survival rate. In parallel with the results of ICP on the water mimosas, a link was also observed between the arsenic accumulation and removal efficiency of arsenic in water. The X-ray spectroscopy (EDX) and scanning electron microscopy (SEM) results showed deformation of water mimosa tissues in response to severe levels of arsenic toxicity. All in all, the results of this investigation suggest that water mimosa can be a reliable phytobioremediator for polluted water with up to 30 ppm concentrations of arsenic. However, further studies are needed on the plant’s genomic and proteomic reactions and other biochemical responses against heavy metals like arsenic. 

## Figures and Tables

**Figure 1 plants-09-01500-f001:**
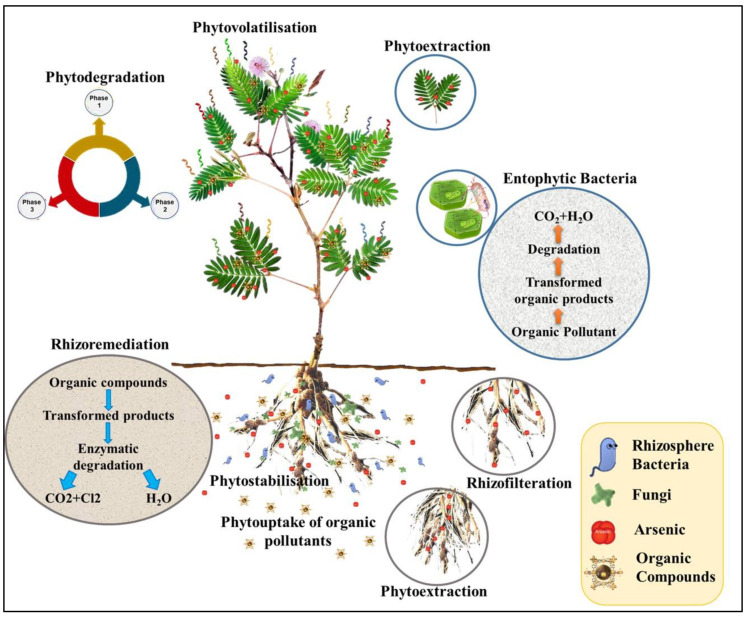
Phytoremediation strategies. The keystone in phytoremediation technologies is contaminants’ interactions with plants’ rhizosphere. Plants absorb metals from the rhizosphere. Roots take up heavy metals through mobilization. Subsequently, the accumulated metals are translocated to the plants’ aerial tissues followed by sequestration in the tissues according to the plants’ tolerance, adapted from [31,33].

**Figure 2 plants-09-01500-f002:**
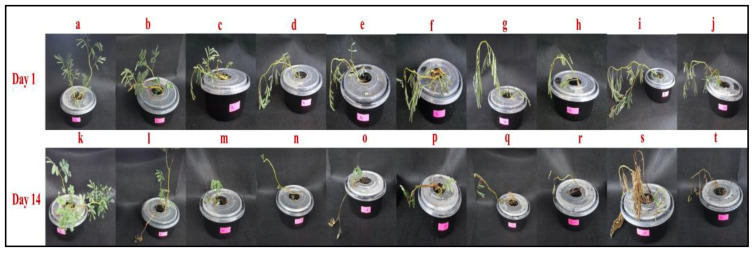
Exposure of water mimosas to different concentrations of arsenic (control, 5, 10, 30, 50, 60, 70, 80, 90, and 100 ppm). A total of 20 tanks were used, and plants were distributed in 3 experimental blocks (10 arsenic concentrations × 3 replications). Each tank (12 × 25 × 10 cm = 3 L) contained a single plant. The control and treated plants were kept at 18–25°C with an 18 h light/6 h dark photoperiod under a light intensity of 500 µmol·m^−2^ for two weeks. The data were collected on the 1st and 14th days. Morphological analysis showed that water mimosas were resistant to low levels of arsenic concentrations (less than 60 ppm). At higher arsenic concentrations, the morphological analysis showed severe symptoms of damage, and ultimately, death of the plant. Collected qualitative and quantitative data confirmed the negative impact of arsenic when its concentrations and time of treatment were increased. Day 1, control = a, 5 pp = b, 10 pp = c, 30 pp = d, 50 ppm = e, 60 ppm = f, 70 ppm = g, 80 ppm = h, 90 ppm = i, and 100 ppm = j. Day 14, control = k, 5 pp = l, 10 pp = m, 30 pp = n, 50 ppm = o, 60 ppm = p, 70 ppm = q, 80 ppm = r, 90 ppm = s, and 100 ppm = t.

**Figure 3 plants-09-01500-f003:**
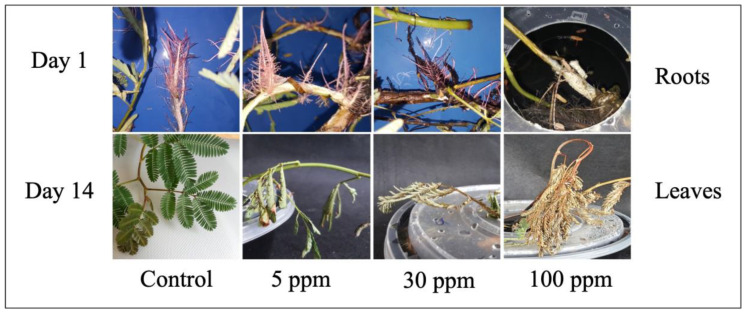
Samples of water mimosas’ leaves and roots under arsenic treatment (control, 5, 30, and 100 ppm) after 1 and 14 days. Each tank (12 × 25 × 10 cm = 3 L) contained a single plant. The control and treated plants were kept at 18–25°C with an 18 h light/6 h dark photoperiod under a light intensity of 500 µmol·m^−2^ for two weeks. Increasing the arsenic levels and the time caused deformation of water mimosa’s parts, necrosis, chlorosis, and yellowing of leaves, as well as root hardening and woody formation.

**Figure 4 plants-09-01500-f004:**
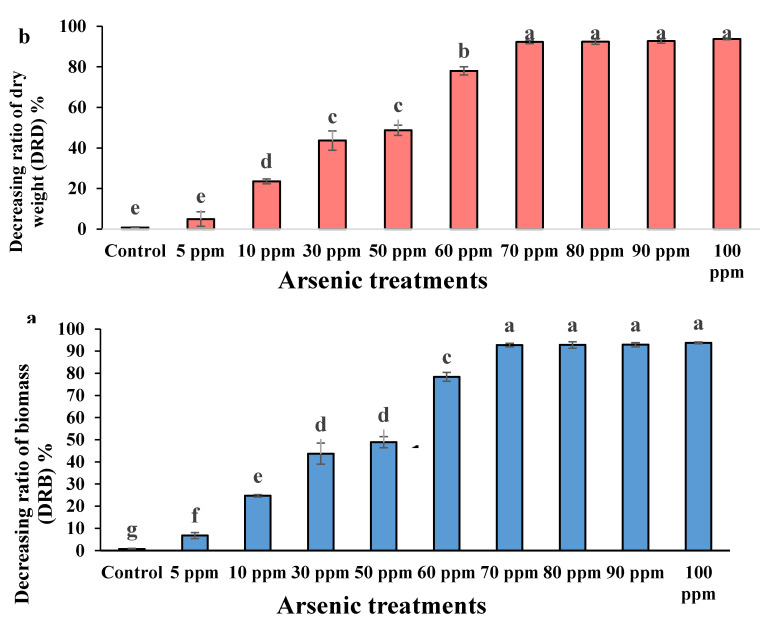
(**a**) Decreasing ratio of biomass (DRB) (%) and (**b**) decreasing ratio of dry weight (DRD) (%) of water mimosas two weeks after treatment in different arsenic concentrations (0, 5, 10, 30, 50, 60, 70, 80, 90, and 100 ppm). Fresh and dry weight of plants were measured on the 1st and 14th days. One-way ANOVA was performed, and bars represent standard errors (SE) of the means of the treatments (*n* = 3) with the same species, if not otherwise stated. Different letters indicate significant differences between arsenic concentrations according to Duncan’s multiple range test (*p* ≤ 0.05).

**Figure 5 plants-09-01500-f005:**
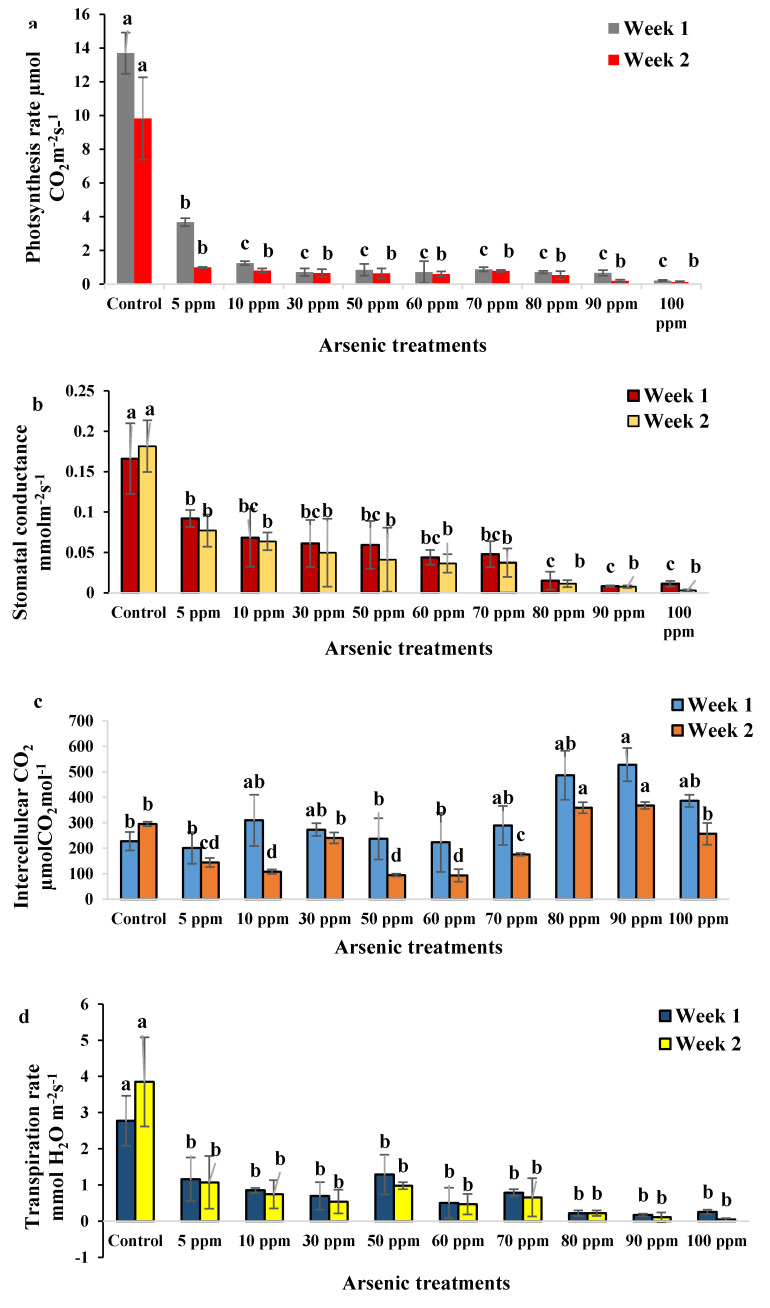
(**a**) Photosynthesis rate (µmol CO_2_ m^−2^·s^−1^); (**b**) Stomatal conductance (mol H_2_O m^−2^·s^−1^); (**c**) Intercellular CO_2_ concentration (µmol CO_2_ mol^−1)^; (**d**) Transpiration rate (mmol H_2_O m^−2^·s^−1^); (**e**) air pressure deficit (kPa); and (**f**) Chlorophyll content (mg/cm^3^), of water mimosas one and two weeks after arsenic treatment at different concentrations (0, 5, 10, 30, 50, 60, 70, 80, 90, and 100 ppm). One-way ANOVA was performed, and bars represent standard errors (SE) of the means of the treatments (*n* = 3) with the same species if not otherwise stated. Different letters indicate significant differences between arsenic concentrations according to Duncan’s multiple range tests (*p* ≤ 0.01). Among the treatments, different letters indicate significant differences according to Duncan’s multiple range test (*p* ≤ 0.05).

**Figure 6 plants-09-01500-f006:**
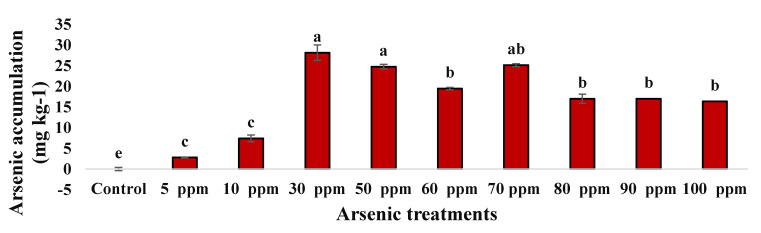
Arsenic accumulation (mg·Kg^−1^) by water mimosas two weeks after arsenic treatments in different concentrations (0, 5, 10, 30, 50, 60, 70, 80, 90, and 100 ppm). One-way ANOVA was performed, and bars represent standard errors (SE) of the means of the treatments (*n* = 3) with the same species if not otherwise stated. Different letters indicate significant differences between arsenic concentrations according to Duncan’s multiple range tests (*p* ≤ 0.01). Among the treatments, different letters indicate significant differences according to Duncan’s multiple range tests (*p* ≤ 0.05).

**Figure 7 plants-09-01500-f007:**
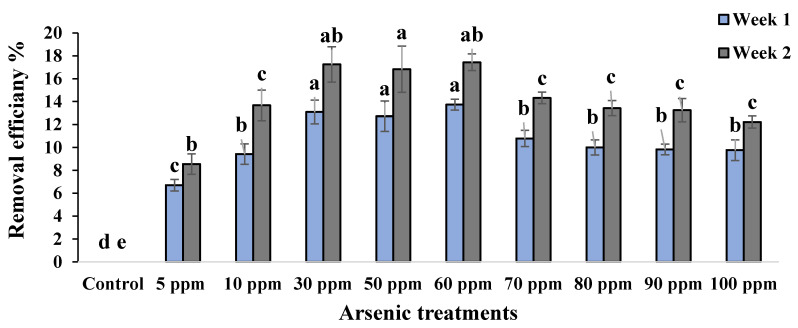
Removal efficiency (%) of water mimosas two weeks after arsenic treatments in different concentrations (0, 5, 10, 30, 50, 60, 70, 80, 90, and 100 ppm). One-way ANOVA was performed, and bars represent standard errors (SE) of the means of the treatments (*n* = 3) with the same species if not otherwise stated. Different letters indicate significant differences between arsenic concentrations according to Duncan’s multiple range tests (*p* ≤ 0.01). Among the treatments, different letters indicate significant differences according to Duncan’s multiple range tests (*p* ≤ 0.05).

**Figure 8 plants-09-01500-f008:**
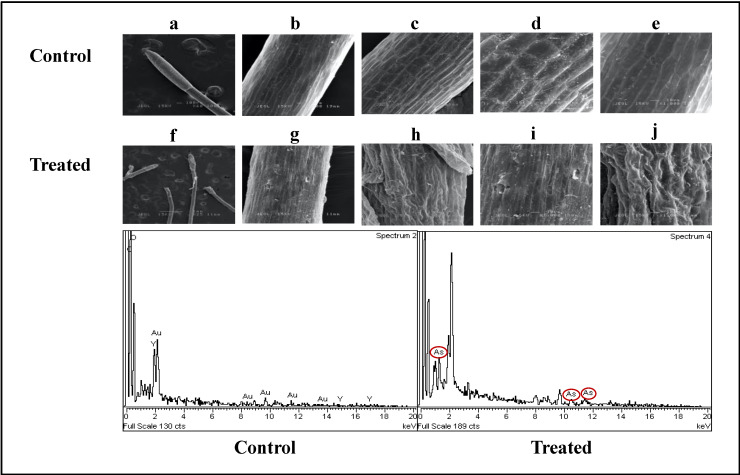
Scanning electron microscopic (SEM) observations were performed on the roots of three independent replications of control and treated (30 ppm sodium heptahydrate arsenate) water mimosa after 14 days of the exposure period. To measure the arsenic contents of root samples, three different spectrums of roots were randomly measured for each image. Comparison of control and treated samples is shown with thinner and younger hairy roots related to the control sample (**a**–**e**) woody, thick, and flaky hairy roots of treated samples (**f**–**j**). (**e**,**j**) show the multi-layered epidermis with the smooth surface in control and rough and scaly in treated samples. Several parenchyma tissues have a rectangular structure in the control samples (**c**–**e**) while in treated one the structure shows irregularity (**f**–**i**). As compared with the control plants (**a**,**b**), the cortex of the treated samples are composed of irregular intercellular nodules on its parenchyma cells (**f**,**g**). Arsenic made visible changes to cell volume (**h**,**j**).

**Table 1 plants-09-01500-t001:** ANOVA results of water mimosas’ decreasing ratio of biomass (DRB) and decreasing ratio of dry weight (DRD) under various concentrations of arsenic treatment.

S.O.V	df	DRB	DRD
**Concentration**	9	0.6241 ^**^	0.66 ^**^
**Replicate**	2	0.001 ^ns^	0.004 ^ns^
**Error**	18	0.0006	0.0043
**Total**	29	-	-
**C.V.**	-	3.08	7.818

S.O.V, source of variation. ^**^ significant at the 0.01 probability levels. ns, non significant. DRB: decreasing ratio of biomass and DRD: decreasing ratio of dry weight (DRD).

**Table 2 plants-09-01500-t002:** Height of frond percentage, green leaves percentage, relative growth rate (RGR) percentage, relative water content (RWC), and tolerance index of water mimosas under different arsenic concentrations.

Arsenic Concentrations (ppm)	Height of Frond (%)	Green Leaves (%)	RGR (g/g·day)	RWC (%)	Ti (%)
**Control**	5 ± 0.2 ^a^	100 ± 0.0 ^a^	0.004 ± 0.00076 ^a^	89.34 ± 1.12 ^d^	100 ± 0.00 ^a^
**5**	5 ± 0.44 ^a^	96 ±1.43 ^b^	0.002 ± 0.00023 ^b^	90.56 ± 1.34 ^cd^	85 ± 1.2 ^b^
**10**	4.5 ± 0.34 ^b^	89 ± 1.76 ^c^	0.001 ± 0.00012 ^bc^	90.67 ± 2.32 ^c^	73 ± 1.5 ^c^
**30**	3.7 ± 0.36 ^c^	71 ± 0.32 ^d^	0.00 ± 0.0 ^d^	91.02 ± 1.12 ^bc^	61± 1.23 ^d^
**50**	2.2 ± 0.24 ^d^	46 ± 0.24 ^e^	0.00 ± 0.0 ^d^	91.03 ± 2.35 ^bc^	47 ± 1.43 ^e^
**60**	1.5 ± 0.12 ^e^	32 ±1.43 ^f^	0.00 ± 0.0 ^d^	91.2 ± 3.23 ^ab^	34 ± 1.84 ^f^
**70**	1.5 ± 0.23 ^e^	29 ± 0.32 ^f^	0.00 ± 0.0 ^d^	91.23 ± 2.22 ^ab^	26 ± 0.98 ^g^
**80**	1.2 ± 0.25 ^e^	7 ± 1.21 ^g^	0.00 ± 0.0 ^d^	91.3 ± 1.33 ^a^	22 ± 1.72 ^h^
**90**	0.9 ± 0.23 ^f^	0.00 ± 0.0 ^e^	0.00 ± 0.0 ^d^	91.32 ± 1.21 ^a^	20 ± 1.32 ^i^
**100**	0.7 ± 0.32 ^g^	0.00 ± 0.0 ^e^	0.00 ± 0.0 ^d^	91.32 ± 1.23 ^a^	16 ± 1.23 ^j^

Plant growth and physiological parameters were measured on the 14th day of experiment. RWC, relative water content; RGR, relative growth rates; and Ti, tolerance index. Different letters indicate significant differences between arsenic concentrations according to Duncan’s multiple range test (*p* ≤ 0.05).

**Table 3 plants-09-01500-t003:** ANOVA results of water mimosas’ chlorophyll content, photosynthesis rate, conductance to H_2_O, intercellular CO_2_ concentrations, transpiration rate, and vapor pressure deficit based on leaf temperature under various concentrations of arsenic treatment.

S.O.V	df	Chlorophyll Content		Photosynthesis Rate		Stomata Conductance		Intercellular CO_2_ Concentrations		Transpiration Rate		Air Pressure Deficit	
		Week 1	Week 2	Week 1	Week 2	Week 1	Week 2	Week 1	Week 2	Week 1	Week 2	Week 1	Week 2
**Concentration**	9	173.6 ^**^	201.7 ^**^	50.53 ^**^	25.82 ^**^	0.0065 ^**^	0.008 ^**^	39,106.1 ^*^	32,985.5 ^**^	1.77 ^**^	3.64 ^**^	0.065 ^ns^	0.188 ^*^
**Replicate**	2	1.12 ^ns^	0.36 ^ns^	0.106 ^ns^	3.45 ^ns^	0.0002 ^ns^	0.001 ^ns^	6335.9 ^ns^	383.87 ^ns^	0.024 ^ns^	0.02 ^ns^	0.048 ^ns^	0.067 ^ns^
**Error**	18	89.53	65.14	0.358	2.99	0.0007	0.001	11,974.96	518.67	0.202	0.372	0.032	0.07184
**Total**	29	-	-	-	-	-	-	-	-	-	-	-	-
**C.V.**	-	11.35	10.41	20.34	19.6	18.22	16.55	34.33	10.47	19.97	20.32	8.698	14.42

S.O.V, source of variation. ^**^ and ^*^ significant at the 0.01 and 0.05 probability levels, respectively. ns, nonsignificant.

**Table 4 plants-09-01500-t004:** Effect of different arsenic concentrations on proline and lipid peroxidation contents of water mimosa.

	MDA Contents (µmol/g FW)		Proline (µmol/g FW)	
Arsenic Concentrations (ppm)	Root	Leave	Root	Leave
**Control**	12.43 ± 1.3 ^g^	15.45 ± 0.81 ^h^	20 ± 0.98 ^fg^	22 ± 0.54 ^h^
**5**	25.54 ± 1.02 ^f^	26.84 ± 1.36 ^g^	23 ± 0.87 ^e^	35 ± 1.39 ^e^
**10**	26.76 ± 0.9 ^e^	28.43 ± 1.21 ^f^	28 ± 0.23 ^d^	37 ± 1.3 ^d^
**30**	32.87 ± 1.2 ^b^	31.34 ± 0.98 ^d^	35 ± 0.4 ^a^	43 ± 1.3 ^a^
**50**	35.47 ± 1.8 ^a^	37.23 ± 0.80 ^a^	33 ± 1.93 ^b^	41 ± 1.74 ^b^
**60**	31.21 ± 1.01 ^c^	34.09 ±0.90 ^b^	30 ± 0.94 ^c^	39 ± 0.87 ^c^
**70**	28.65 ± 0.87 ^d^	33.35 ± 1.89 ^c^	27 ± 0.76 ^d^	28 ± 0.36 ^f^
**80**	26.67 ± 0.67 ^e^	30.12 ± 1.24 ^e^	21 ± 0.89 ^f^	25 ± 0.87 ^g^
**90**	20.76 ± 0.56 ^h^	24.34 ± 1.78 ^i^	19 ± 0.4 ^g^	22 ± 0.87 ^h^
**100**	18.23 ± 0.76 ^i^	24.12 ± 1.23 ^i^	19 ± 0.5 ^g^	18 ± 0.45 ^i^

Proline and lipid peroxidation contents were measured on the 14th day of experiment. MDA, lipid peroxidation. Different letters indicate significant differences between arsenic concentrations according to Duncan’s multiple range tests (*p* ≤ 0.05).

**Table 5 plants-09-01500-t005:** ANOVA of arsenic accumulation (mg·Kg^−1^) and removal efficiency (ppm) of water mimosa after arsenic treatment.

S.O.V	df	ICP Water Mimosa	Removal Efficiency after 7 Days	Removal Efficiency after 14 Days
Concentrations	9	277.330 ^**^	47.31 ^**^	81.06 ^**^
Replicate	2	1.303 ^ns^	0.709 ^ns^	0.750 ^ns^
Error	18	0.7125	0.661	1.46
Total	29	-	-	-
C.V.	-	5.290	8.419	9.54

S.O.V, source of variation. ^**^ significant at the 0.01 probability levels. ns, nonsignificant.

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
