# Peer review of "Assessment of Water Mimosa (Neptunia oleracea Lour.) Morphological, Physiological, and Removal Efficiency for Phytoremediation of Arsenic-Polluted Water"

_plants, 2020, doi:10.3390/plants9111500_

Round 1
Reviewer 1 Report
Arsenic is a worldwide environmental problem, with more than 150 million people affected by arsenic-contaminated water. The people live in South and Southeast Asia drink water up to 1000 µg L‑1 arsenic, which is a level 100 times higher than arsenic drinking water guideline value (10 µg/L) (WHO, 2011). Water mimosa (Neptunia oleracea) has been widely identified as a feasible phytoremediator to clean up aquatic systems. However the phytoremediation potential of this species to various concentrations arsenic has not been reported. The submitted manuscript concerns a number of plant physiological and growth responses indicating that water mimosa is able to tolerate up to 30 mg L-1 when the plants are exposed to arsenic for 14 days. Moreover the highest accumulation of arsenic in roots as well as the removal efficiency of plants were recorded at the same concentration of metalloid. According to presented data mimosa reveals considerable phytoremediation capabilities and is especially applicable for phytofiltration of arsenic-contaminated water.
Abstract
The results relating to the accumulation of arsenic in mimosa plants and the removal efficiency of this metalloid from water should be mentioned in the abstract because these outcomes, in fact, lead to the conclusion that “water mimosa is a reliable bioremediator for removing arsenic from aquatic systems” (line 30 and 31).
Introduction
Line 58-63 Water mimosa is a phytoremediator dedicated to clean up water bodies. What is the concentration of arsenic in Malaysia's water resources? This information would be more valued than the data concerning the contamination of arsenic in the soil.
Line 64-71 This paragraph should be shortened as there is no need to characterize in detail the techniques, which are “expensive and needs long-time maintenance”.
Line 72-114 This part of the text should be more concise. It should focus on the phytofiltration and the potential application of mimosa to the removal of contaminants from polluted water.
Line 105-107 “Reportedly, N. oleracea has been broadly applied for the decontamination or reduction of contaminant levels in waters surrounding some Asian countries including Malaysia, Thailand, Indonesia, Philippines, and Vietnam (Ab Wahab et al., 2016).”
There isn’t any information about N. oleracea in the article of Ab Wahab et al., 2016.
Line 115- 17 The main objective of the study should also include the efficiency assessment of water mimosa for phytoremediation of arsenic-polluted water.
Results (figures) and discussion
- Figures:
1.1.The quality of the figures 1, 2 and 3 should be improved as they are blurred and poorly legible.
1.2.Every caption should provide sufficient information to the readers without referring to the related text in the manuscript. It should include information on important aspects such as description (the type of experiment), method or technique used (sample size, type of plant model used, metal concentration, duration of the experiment etc.), and the result obtained (p-value).
1.3.The caption should contain the units of measurement the same as the unites presented on the graphs (Fig. 4, 5, 6, 7).
- Results
2.1. Line 124-126 “Quantitative and qualitative observations of the plants at the 1 st , 3 rd, 7 th, 10 th , and 14 th days showed that the growth and reactions of the water mimosas varied in response to the different arsenic treatment concentrations (Figure 2).”
Why plants at the 3 rd and 10 th days are not shown in the Figure 2, despite they are mentioned in the text?
2.2.What was the reason that some parameters such as: decreasing ratio of biomass (DRB), decreasing ratio of dry weight (DRD) (Fig. 4), the height of frond percentage, green leaves percentage, relative growth rate (RGR) percentage, relative water content (RWC), and tolerance index (Ti) (Tab. 2), proline and MDA contents (Tab. 4), arsenic accumulation (Fig. 7) were estimated only after 14 days of exposure to different arsenic concentrations? On the other hand photosynthesis rate, stomata conductance, intercellular CO2 concentration, transpiration rate air pressure deficit and chlorophyll content (Fig. 5) as well as removal efficiency were assessed after 7 and 14 days of arsenic treatment.
2.3.Why the impact of arsenic on histology of mimosa was observed in the roots of plants treated with 50 mg L-1 of arsenic for 14 days (Figure 8), whereas the highest accumulation of metalloid was recorded at 30 mg L-1 (Figure 6)?
- Discussion
Line 172 “The developed vacuoles transport water through the cell wall using aquaporin as an advanced transport channel (Toyota et al., 2018).”
Aquaporins activity regulates transmembrane water transport.
Line 393-395 “The increased rate of MDA in the current study could be due to the hyperactivity of different antioxidative enzymes in the defence mechanisms and reduced H2O2 levels.”
Line 514-514“Observations on the proline and lipid peroxidation contents showed water mimosas’ adequate tolerance by inducing the mentioned antioxidant under a moderate level of arsenic treatment (30 mgL-1)”.
This conclusions are false.
According to Zhan et al. 2007:
1)“Malondialdehyde (MDA) is a cytotoxic product of lipid peroxidation and an indicator of free radical production and consequent tissue damage.” (page 45, quoted after (Ohkawa et al., 1979)
2)“In this study, increased free radical generation was found in two mangrove plants under heavy metal stress as indicated by the MDA production…” (page 48)
3)“We think that the reduction of MDA concentration was due to increased antioxidative enzyme activities, which reduced H2O2 levels and membrane damage.” (page 49)
Line 376- 378 “The MDA of the water mimosas spiked along with increases in arsenic concentrations up to 50 mg L-1. However, at arsenic concentrations of more than 50 mg L-1, the level of MDA decreased significantly (Table 4).”
These results suggest that water mimosa was able to tolerate arsenic up to 30-50 mg L-1, in spite of the fact, that lipid peroxidation was enhanced by this metalloid and, as a consequence cell membrane stability was affected. The arsenic-dependent decrease in MDA accumulation at the range of concentration 60-100 mg L-1 might be attributed to the reduction in survival rate of plants.
References should be completed e. g. Prum et al., 2018
Author Response
Reviewer 1
Comments and Suggestions for Authors
Arsenic is a worldwide environmental problem, with more than 150 million people affected by arsenic-contaminated water. The people live in South and Southeast Asia drink water up to 1000 µg L‑1 arsenic, which is a level 100 times higher than arsenic drinking water guideline value (10 µg/L) (WHO, 2011). Water mimosa (Neptunia oleracea) has been widely identified as a feasible phytoremediator to clean up aquatic systems. However the phytoremediation potential of this species to various concentrations arsenic has not been reported. The submitted manuscript concerns a number of plant physiological and growth responses indicating that water mimosa is able to tolerate up to 30 ppm when the plants are exposed to arsenic for 14 days. Moreover the highest accumulation of arsenic in roots as well as the removal efficiency of plants were recorded at the same concentration of metalloid. According to presented data mimosa reveals considerable phytoremediation capabilities and is especially applicable for phytofiltration of arsenic-contaminated water.
Abstract
Comment 1:
The results relating to the accumulation of arsenic in mimosa plants and the removal efficiency of this metalloid from water should be mentioned in the abstract because these outcomes, in fact, lead to the conclusion that “water mimosa is a reliable bioremediator for removing arsenic from aquatic systems” (line 30 and 31).
Answer:
Many thanks for the comment. The following statement has now been added to the abstract:
“In addition, the highest arsenic accumulation and arsenic removal efficacy were observed at the range of 30-60 ppm”
Introduction
Comment 2:
Line 58-63 Water mimosa is a phytoremediator dedicated to clean up water bodies. What is the concentration of arsenic in Malaysia's water resources? This information would be more valued than the data concerning the contamination of arsenic in the soil.
Answer:
Thank you. We have agreed to change the statement and added new information on the arsenic level from soil to water. Based on our literature, arsenic is a site-specific and time-dependent pollution and the arsenic pollution levels vary in different natural water resources in Malaysia. On the other hand, the recorded levels of arsenic in different rivers of Malaysia were between 2.00 to 54.00, which was cited by Sobahan et al. (2013) in the Selangor River basin in Malaysia. The following sentences are new data of arsenic concentrations in rivers of Malaysia and also the international environment guidelines which are below 10 μg/L for drinking water samples.
After editing:
“It has been reported that the concentration of arsenic in rivers across Malaysia is between 2.00 to 54.00 μg/L (Sobahan et al., 2013; Ab Razak et al., 2015; Sakai et al., 2017; Othman et al., 2018; Hwi et al., 2020). This is a concern as it exceeds international environment guidelines which is below 10 μg/L for drinking water samples (Fernandez-Luqueno et al., 2013).”
Comment 3:
Line 64-71 This paragraph should be shortened as there is no need to characterize in detail the techniques, which are “expensive and needs long-time maintenance”.
Answer:
The paragraph has been shortened now to:
After editing:
Until now, several treatment technologies have been announced for the removal of arsenic from water bodies (AlJaberi 2018). However, it is worth mentioning that the use of the most effective technology depends on plenty of factors such as environmental impact, operational cost and capital investment, the initial metal concentration, and plant reliability and flexibility, etc. (Fu and Wang., 2011).
Comment 4:
Line 72-114 This part of the text should be more concise. It should focus on the phytofiltration and the potential application of mimosa to the removal of contaminants from polluted water.
Answer:
The paragraph has been changed as follows:
After editing:
For the past two decades, phytoremediation has been developed as a green, non-invasive and economic alternative to different conventional civil engineering-based strategies for the remediation of water, soil and even residences polluted with heavy metals (Wei et al., 2019; Prabakaran et al., 2019).
Strategies employed under phytoremediation include phytodegradation (employing plant or microorganism to degrade contaminants) (Arabnezhad et al., 2019), phytoaccumulation (employing algae or plant to accumulate contaminants in their areal parts) (Nasr., 2020), phytostabilisation (employing plant to reduce the heavy metal mobility in soil) (Castaldi et al., 2018), phytofiltration (employing plant biomass and their associated rhizospheric microorganisms to refine contaminants) (Nasr 2020), phytovolatilisation (employing plant to absorb contaminants and transpire them into the atmosphere in the volatile shape) (Ossai et al., 2020), and Rhizodegradation (employing plant to degrade contaminants using rhizosphere microbes’ mediation ( Yadav et al., 2018; Prabakaran et al., 2019; Hoang et al., 2020) (Figure 1).
Comment 5:
Line 105-107 “Reportedly, N. oleracea has been broadly applied for the decontamination or reduction of contaminant levels in waters surrounding some Asian countries including Malaysia, Thailand, Indonesia, Philippines, and Vietnam (Ab Wahab et al., 2016).”
There isn’t any information about N. oleracea in the article of Ab Wahab et al., 2016.
Answer:
Many thanks for the comment. After we checked the references we notified that the above sentences are related to (Wahab et al., 2014). We updated the reference in the text from (Ab Wahab et al., 2016) to (Wahab et al., 2014).
Background: Neptunia oleracea or water mimosa has been extensively used as a water treatment agent in Asian countries such as Thailand, Vietnam, Philippines, Indonesia and Malaysia.
Wahab, A., Ismail, S. S., Abidin, E. Z., & Praveena, S. (2014). Neptunia oleracea (water mimosa) as phytoremediation plant and the risk to human health: A review. Adv Environ Biol, 8, 187-194.
Comment 6:
Line 115- 17 The main objective of the study should also include the efficiency assessment of water mimosa for phytoremediation of arsenic-polluted water.
Answer:
Thank you. We have added the following statement in the draft.
“In addition, the present investigation aims to get insights into the removal efficiency of water mimosa for phytoremediation of arsenic-polluted water.”
Results (figures) and discussion
Comment 7:
- Figures: 1.1.The quality of the figures 1, 2 and 3 should be improved as they are blurred and poorly legible.
Answer:
Thank you. The blurred figures have been deleted and replaced with the high resolution ones.
Comment 8:
1.2. Every caption should provide sufficient information to the readers without referring to the related text in the manuscript. It should include information on important aspects such as description (the type of experiment), method or technique used (sample size, type of plant model used, metal concentration, duration of the experiment etc.), and the result obtained (p-value).
Answer:
Many thanks, we have improved the caption of figures by adding the information regarding the implementation of the experiment as follow:
After editing:
Figure 2. Exposure of water mimosas to different concentrations of arsenic (control, 5 ppm, 10 ppm, 30 ppm, 50 ppm, 60 ppm, 70 ppm, 80 ppm, 90 ppm, and 100 ppm). A total of 30 tanks were used and plants were distributed in 3 experimental blocks (10 arsenic concentrations × 3 replications). Each tank (12 × 25 × 10 cm = 3 L) contained a single plant. The control and treated plants were kept at 18-25 °C with an 18 h light/6 h dark photoperiod under a light intensity of 500 µmol.m−2 for two weeks. The data were collected on the 1st and 14th days. Morphological analysis showed that water mimosas were resistant to low levels of arsenic concentrations (less than 60 ppm). At higher arsenic concentrations, the morphological analysis showed severe symptoms of damages, and ultimately, death of the plant. Collected qualitative and quantitative data confirmed the negative impact of arsenic when increasing its concentrations and time of treatment.
Figure 3. Samples of water mimosas’ leaves and roots under arsenic treatment (control, 5 ppm, 50 ppm, and 100 ppm) on days 7 and 14. Each tank (12 × 25 × 10 cm = 3 L) contained a single plant. The control and treated plants were kept at 18-25 °C with an 18 h light/6 h dark photoperiod under a light intensity of 500 µmol.m−2 for two weeks. Increasing the arsenic levels and the time caused deformation of water mimosa’s parts, necrosis, chlorosis, and yellowing of leaves as well as root hardening and woody formation.
Comment 9:
1.3. The caption should contain the units of measurement the same as the unites presented on the graphs (Fig. 4, 5, 6, 7).
Answer:
Many thanks for the comments. We have added the units and the type of statistical analysis (one-way ANOVA) in the figure captions. The highlighted parts represent the added sections.
Figure 4. (a) Decreasing ratio of biomass (DRB) (%) and (b) decreasing ratio of dry weight (DRD) (%) of water mimosas two weeks after treatment in different arsenic concentrations (0, 5, 10, 30, 50, 60, 70, 80, 90, and 100 ppm). Fresh and dry weight of plants were measured at the 1st and 14th days. One- way ANOVA was performed, and bars represent standard errors (SE) of the means of the treatments (n = 3) with the same species if not otherwise stated. Different letters indicate significant differences between arsenic concentrations according to Duncan’s multiple range tests (p ≤ 0.05).
Figure 6. Arsenic accumulation (mg Kg-1) by water mimosas two weeks after arsenic treatments in different concentrations (0, 5, 10, 30, 50, 60, 70, 80, 90, and 100 ppm). One-way ANOVA was performed, and bars represent standard errors (SE) of the means of the treatments (n = 3) with the same species if not otherwise stated. Different letters indicate significant differences between arsenic concentrations according to Duncan’s multiple range tests (p ≤ 0.01). Among the treatments, different letters indicate significant differences according to Duncan’s multiple range tests (p ≤ 0.05).
Figure 7. Removal efficiency (%) of water mimosas two weeks after arsenic treatments in different concentrations (0, 5, 10, 30, 50, 60, 70, 80, 90, and 100 ppm). One- way ANOVA was performed, and bars represent standard errors (SE) of the means of the treatments (n = 3) with the same species if not otherwise stated. Different letters indicate significant differences between arsenic concentrations according to Duncan’s multiple range tests (p ≤ 0.01). Among the treatments, different letters indicate significant differences according to Duncan’s multiple range tests (p ≤ 0.05).
Figure 5. (a) Photosynthesis rate (μmolCO2m-2s-1), (b) stomata conductance (molH2Om-2 s-1), (c) intercellular CO2 concentration (μmolCO2mol-1), (d) transpiration rate (mmolH2Om-2 s-1), (e) air pressure deficit (kPa), and (f) chlorophyll content (mgcm3) of water mimosas one and two weeks after arsenic treatment at different concentrations (0, 5 , 10 , 30 , 50 , 60 , 70 , 80 , 90 , and 100 ppm). One-way ANOVA was performed, and bars represent standard errors (SE) of the means of the treatments (n = 3) with the same species if not otherwise stated. Different letters indicate significant differences between arsenic concentrations according to Duncan’s multiple range tests (p ≤ 0.01). Among the treatments, different letters indicate significant differences according to Duncan’s multiple range tests (p ≤ 0.05).
All in all, based on the journal format (published draft sample as follow) and with all due respect to the reviewer comments, we noticed that there is no need to explain the graph results in the captions e.g. “Saleem, M. H., Ali, S., Kamran, M., Iqbal, N., Azeem, M., Tariq Javed, M., ... & Alkahtani, S. (2020). Ethylenediaminetetraacetic Acid (EDTA) Mitigates the Toxic Effect of Excessive Copper Concentrations on Growth, Gaseous Exchange and Chloroplast Ultrastructure of Corchorus capsularis L. and Improves Copper Accumulation Capabilities. Plants, 9(6), 756.”
- Results
Comment 10:
2.1. Line 124-126 “Quantitative and qualitative observations of the plants at the 1 st , 3 rd, 7 th, 10 th , and 14 th days showed that the growth and reactions of the water mimosas varied in response to the different arsenic treatment concentrations (Figure 2).”
Why plants at the 3 rd and 10 th days are not shown in Figure 2, despite they are mentioned in the text?
Answer:
Thank you for the important point of view. The reviewer mentioned correctly, so as we specifically aimed to compare the arsenic effect at the first and last days of treatment, in this regard, we decided to focus on slicking up the high resolutions pictures related to 1st and 14th days. By the way, we deleted the information regarding the other days in the material and methods and relevant content.
Comment 11:
2.2.What was the reason that some parameters such as: decreasing ratio of biomass (DRB), decreasing ratio of dry weight (DRD) (Fig. 4), the height of frond percentage, green leaves percentage, relative growth rate (RGR) percentage, relative water content (RWC), and tolerance index (Ti) (Tab. 2), proline and MDA contents (Tab. 4), arsenic accumulation (Fig. 7) were estimated only after 14 days of exposure to different arsenic concentrations? On the other hand photosynthesis rate, stomata conductance, intercellular CO2 concentration, transpiration rate air pressure deficit and chlorophyll content (Fig. 5) as well as removal efficiency were assessed after 7 and 14 days of arsenic treatment.
Answer:
The Gas Exchange Attributes (photosynthesis rate, stomatal conductance, intercellular CO2 concentration, transpiration rate air pressure deficit and chlorophyll content), which were estimated after 7 and 14 days, were non-destructive traits and we measured them by LiCor 6400 Portable Photosynthesis System mechanic. On the other hand, the measurement method of the majority of the other traits was destructive and if we did it at different times we would have destroyed (inaccurate or misleading results) the part of the plant during the experiment which would introduce some technical errors. However, there were some other traits which we could measure them by non-destructive method, but to have the uniform data, the table and/or figures were presented the same trend data; which is day 14. On the other hand, arsenic accumulation in plants was also measured after 14 days as this assessment of this trait is also a part of destructive methods. Regarding removal efficiency, as the samples were water and we took the same amount of water from all the tanks, simultaneously, we could measure it at different times. However, to avoid error due to sampling water, we did it just twice.
Comment 12:
2.3. Why the impact of arsenic on histology of mimosa was observed in the roots of plants treated with 50 mg L-1 of arsenic for 14 days (Figure 8), whereas the highest accumulation of metalloid was recorded at 30 mg L-1 (Figure 6)?
Answer:
Thank you. As the reviewer mentioned correctly, the treatment of 30 ppm arsenic was better and definitely our choice. Although, we used EDX and SEM for the mentioned treatment, however, during the editing (maybe) we wrote 50 ppm, mistakenly. We hereby announced that the picture is for 30 ppm, and we have corrected it in the draft accordingly.
- Discussion
Comment 13:
Line 172 “The developed vacuoles transport water through the cell wall using aquaporin as an advanced transport channel (Toyota et al., 2018).”
Aquaporins activity regulates transmembrane water transport.
Answer:
Thank you. We have rewritten in a correct statement as follow:
Water flux across the tonoplast/plasma membrane, and the parallel leaves reaction are relevant to the water channel aquaporin as a particular membrane protein (Hagihara and Toyota., 2020).
Hagihara, T., & Toyota, M. (2020). Mechanical Signaling in the Sensitive Plant Mimosa pudica L. Plants, 9(5), 587.
Comment 14:
Line 393-395 “The increased rate of MDA in the current study could be due to the hyperactivity of different antioxidative enzymes in the defence mechanisms and reduced H2O2 levels.”
Line 514-514“Observations on the proline and lipid peroxidation contents showed water mimosas’ adequate tolerance by inducing the mentioned antioxidant under a moderate level of arsenic treatment (30 mgL-1)”.
This conclusions are false.
According to Zhan et al. 2007:
1)“Malondialdehyde (MDA) is a cytotoxic product of lipid peroxidation and an indicator of free radical production and consequent tissue damage.” (page 45, quoted after (Ohkawa et al., 1979)
2)“In this study, increased free radical generation was found in two mangrove plants under heavy metal stress as indicated by the MDA production…” (page 48)
3)“We think that the reduction of MDA concentration was due to increased antioxidative enzyme activities, which reduced H2O2 levels and membrane damage.” (page 49)
Answer:
Many thanks for your constructive comments. We have revise the conclusion and the correct statement is as follow:
The MDA content could increase with heavy metal stress due to the concentration-dependent free radical production. The better the oxidative damage protection is, the more quickly the antioxidative system could be to up-regulate. Peroxidising activity could lead to the MDA Elimination. Increasing the antioxidative enzyme activities such as peroxidising activity lead to the MDA Elimination, subsequently, reduction of H2O2 amounts and membrane damage (Zhan et al. 2007).
We have corrected the statement as follow:
Observations on the antioxidant content showed water mimosas’ adequate tolerance under 30 ppm of arsenic treatment for proline.
Comment 15:
Line 376- 378 “The MDA of the water mimosas spiked along with increases in arsenic concentrations up to 50 mg L-1. However, at arsenic concentrations of more than 50 mg L-1, the level of MDA decreased significantly (Table 4).”
These results suggest that water mimosa was able to tolerate arsenic up to 30-50 mg L-1, despite the fact, that lipid peroxidation was enhanced by this metalloid and, as a consequence cell membrane stability was affected. The arsenic-dependent decrease in MDA accumulation at the range of concentration 60-100 mg L-1 might be attributed to the reduction in the survival rate of plants.
Answer:
Many thanks for the important point. We have rearranged the mentioned section as followed:
Results of MDA and proline section:
“Analysis of the water mimosas under different arsenic treatments at day 14 showed various proline and MDA (lipid peroxidation) contents. The highest level of MDA was observed in 50 ppm at 35.47± 1.8 and 37.23± 0.80 (µmol/g FW) for root and leaves, respectively (Table 4). Additionally, the level of MDA in the leaves was higher than in the roots under all arsenic treatments (Table 4). Subsequently, the higher level of proline was observed in the roots (35±0.4 µmol/g FW) and leaves (43±1.3 µmol/g FW) of the sample treated with 30 ppm arsenic (Table 4). In the treated root samples, the lowest levels of proline were observed for plants treated with 90 ppm (19±0.4 µmol/g FW) and 100 ppm (19±0.5 µmol/g FW) arsenic (Table 4).”
Discussion of MDA and proline section:
These results showed that water mimosa was able to tolerate arsenic up to 30-50 ppm (Table 4), even though lipid peroxidation was enhanced by this metalloid and, as a consequence, cell membrane stability was affected (Ahmadi Mousavi et al., 2009; Monem et al., 2012). The arsenic-dependent decrease in MDA accumulation at the range of 60-100 ppm might be attributed to the reduction in the survival rate of plants.
On the other hand, we noticed that we cited 50 ppm arsenic as the most effective level of arsenic for proline contents, however, the most effective level was 30 ppm and we did amendment, accordingly.
Comment 16:
References should be completed e. g. Prum et al., 2018
Answer:
Many thanks, the mentioned reference is added and all the references double-checked in the content and bibliography.

Reviewer 2 Report
This MS is mainly about the description of the morphological and physiological changes in Water Mimosa treated with different amount of arsenic.
The MS is descriptive and contains some major and minor issues:
Major issues:
• There are a lot of different concentrations used, but why the 20 and 40 ppm were skipped? Are there any data regarding this concentration or what was the reason to skip them?
• How were the plants selected for the experiments? Were there any standardization or they were used just as they were found in the pond? Plants in different growth phases can react to HMs differently, so it very important in an experiment like this.
• It is stated that quantitative observations were done regarding the growth parameters (see line 124), it would be nice to see the real effects, not only the percentages (like in the table 2.).
• How was the DRB calculated in the case of control plants? If it was compared to the control plants, how can we see any change?
• It would be also nice to see the As contents in the root and shoot separately, not only on whole-plant level.
• What the different colours mean in the case of fig 7? There were two data sets visualized on the same graph but there isn’t any information regarding it.
• What was the reason for choosing the 50 ppm treatment for the EDX and SEM experiments? For me, it seems that the 30 ppm could be a better choice.
• I think that the whole statistical analysis should be redone, or at least in case of table 1 and 3. I do not see any reason to compare the different concentration in this way, it not informative. We can get much more information from a different type of tables (with more data, like in the case of table 2).
• I miss from the final conclusion the fact that where and how can we use these plants for phytoremediation? As we can see probably not every environment (based on the concentration of As) is suitable for Water Mimosa plants.
Minor issues:
• There are some misinterpreted statements. See line 172: ‘The developed vacuoles transport water through the cell wall using aquaporin as an advanced transport channel (Toyota et al., 2018). The functions of these channels are similar to those of ion channels.’ These sentences should be reconsidered. The vacuoles itself cannot transport water through the cell walls, the accumulation or release of ions in these vacuoles can help in the regulation of water transport through the plasma membrane with the help of aquaporins (which are very different from other ion channels, in the perspective of the function), not through the cell walls.
• The units of measure should be standardized also. PPM is used on graphs while mg/L is used in the text.
• There isn't any mention about the time period of the treatment in the description of table 2.
Author Response
Reviewer 2
This MS is mainly about the description of the morphological and physiological changes in Water Mimosa treated with different amount of arsenic. The MS is descriptive and contains some major and minor issues:
Major issues:
Comment 1:
- There are a lot of different concentrations used, but why the 20 and 40 ppm were skipped? Are there any data regarding this concentration or what was the reason to skip them?
Answer:
Thank you for the comment. We planned to arrange 10 levels of arsenic contamination (treated and control) focusing on higher concentrations rather than lower concentrations. Besides, the severe effects of arsenic were more taken into consideration than the slight effects. We have assessed all the concentrations including 20ppm and 40ppm, however, according to the final results of all treatments, and the aim of this study, we decided to put the selected concentrations.
Comment 2:
- How were the plants selected for the experiments? Were there any standardization or they were used just as they were found in the pond? Plants in different growth phases can react to HMs differently, so it very important in an experiment like this.
Answer:
Thank you so much for the important point. Initially the plants were collected from a natural pound inside the university. We collected the plants with the same size and weight. We acclimatized them for one month, and after that, new shoots were collected and transferred to a greenhouse under controlled conditions. Therefore they were of the same age and almost the same morphology. Nonetheless, we have to mention that, the number of leaves was not similar in this study, as it was difficult to control the leaves response to touch and other environmental stimuli. However, in the manuscript we just mentioned the collection and acclimatization phase. All in all, we added the “similar size and weight” term in the draft as followed:
Local naturally-grown N. oleracea aquatic plants with similar weight and size were collected from Universiti Putra Malaysia’s pond (2°59'23.8"N, 101°42'46.5"E), and the plant species was confirmed by the Biodiversity Unit (UBD) of Institute of Bioscience, Universiti Putra Malaysia. The collected plants were acclimatized under hydroponic conditions in tanks (12 × 25 × 10 cm = 3 L) containing 0.20x of Hoagland solution with an aeration system.
Additionally, we followed this references to collect the plants from the pound in the first phase of the project: Wahab, A. S. A., Ismail, S. N. S., Praveena, S. M., & Awang, S. (2014). Heavy metals uptake of water mimosa (Neptunia oleracea) and its safety for human consumption. Iranian Journal of Public Health, 43(Supple 3), 103-111.
Comment 3:
- It is stated that quantitative observations were done regarding the growth parameters (see line 124), it would be nice to see the real effects, not only the percentages (like in the table 2.).
Answer:
Thank you. As the plants were not 100% similar and the morphological traits such as height, weight and number of leaves vary (slightly) between plants, we decided to calculate the ratio to increase the accuracy of the experiment. In addition, for some morpho- and physiological experiments in the nursery or greenhouse, evaluation of PGR, RWC and Ti showed promising results.
Comment 4:
- How was the DRB calculated in the case of control plants? If it was compared to the control plants, how can we see any change?
Answer:
Thank you for the comment. The raw data of biomass weight related to control plants on day 1 for all three biological replications were 189.9, 189.4 and 188.9, whereas these amounts at day 14 for the same plants were 189.0, 188.7, and 188.4. As we can see, the amounts of changes were decreased for each replication, for example, replicate 1 (189.9 to 189.0) replicate 2 (189.4 to 188.7), and replicate 3 (188.9 to 188.4), and the ratio of the DRB averages was less than 1.00%. Our statistical analysis results showed non-significant differences between the recorded data at days 1 and 14.
Based on our literature, we interpreted this decreasing as the reason for the insufficient nutrients in the distilled water. However, it was non-significant changes because the duration of the experiment was short and if we continued the experiment more than 14 days, maybe we recorded even the significant changes between days 1 and 14 for control plants, as well. Published reports on N. oleracea and N. plena have shown that they fix their nitrogen via a symbiotic relationship with soil and bacteria stored in specialised root nodules. N. oleraceae is nodulated by Allorhizobium undicola (De Lajudie et al. 1998) and Devosia Neptuniae (Rivas et al. 2002; Rivas et al. 2003).
Hereupon, the following sentences has been added “Compared to aboveground parts (leaves and stems), heavy metals could be accumulated in the root of N. oleracea (Mishra et al., 2008a). The higher accumulation of heavy metals indicated the removal efficiency of this plant through rhizofiltration process (Wahab et al., 2014).”
Comment 5:
- It would be also nice to see the As contents in the root and shoot separately, not only on the whole-plant level.
Answer:
Many thanks for the valuable comment. We measured the arsenic level in just root and not the whole plant or shoot. As has been mentioned in our literature review, reports have shown that there were non-significant differences for heavy metals i.e. lead, copper and cadmium contents in the areal part of water mimosa and water hyacinth, therefore we just recorded arsenic content in the root (syuhaida et al., 2014).
Syuhaida, A. W. A., Norkhadijah, S. I. S., Praveena, S. M., & Suriyani, A. (2014). The comparison of phytoremediation abilities of water mimosa and water hyacinth.
Comment 6:
- What the different colours mean in the case of fig 7? There were two data sets visualized on the same graph but there isn’t any information regarding it.
Answer:
Many thanks, the mentioned fig is replaced with the correct style.
Comment 7:
- What was the reason for choosing the 50 ppm treatment for the EDX and SEM experiments? For me, it seems that the 30 ppm could be a better choice.
Answer:
Thank you for your attention. As the reviewer mentioned correctly, the treatment of 30 ppm arsenic was better and definitely our choice. Although, we used EDX and SEM for the mentioned treatment, however, during the editing (maybe) we wrote 50 ppm, mistakenly. We hereby announced that the picture is for 30 ppm, and we have corrected it in the draft accordingly.
Comment 8:
- I think that the whole statistical analysis should be redone, or at least in case of table 1 and 3. I do not see any reason to compare the different concentration in this way, it not informative. We can get much more information from a different type of tables (with more data, like in the case of table 2).
Answer:
Thank you so much for the comment, however we have to mention that to analyze the quantitative data, from a statistical point of view, first, we need to apply ANOVA test. The one-way analysis of variance (ANOVA) is used to determine whether there are any statistically significant differences between the means of two or more independent (unrelated) groups (although you tend to only see it used when there are a minimum of three, rather than two groups). Therefore, we applied ANOVA test to see the differences between the treatments (table 1 and 2). These two tables have confirmed the significant differences between the means. When we found that there is some level of significance between treatment, then we stepped forward to find the differences between the specific mean (table 2) using different means of comparison methods (a consideration or estimate of the similarities or dissimilarities between two treatments). For example, in table one we identified that there is some level of significance between the treatments for DRB and DRD traits. In the second step, we showed the mean of comparison for the mean of traits in different concentrations of arsenic (Figure 4). Additionally, in table 3 we showed the significant differences of mentioned traits, and in the second phase, we demonstrated the differences between each mean of treatments, separately. Although the application of ANOVA is just limited to the identification of differences for each trait, we can delete the mentioned tables. But, as we mentioned earlier, from a statistical point of view, to go to the second step needs to evaluate the first step. With all of our due respect, we think it’s necessary to have both tables.
Comment 9:
- I miss from the final conclusion the fact that where and how can we use these plants for phytoremediation? As we can see probably not every environment (based on the concentration of As) is suitable for Water Mimosa plants.
Answer:
Thank you. We have rephrased the sentence as below:
“All in all, the result of this investigation suggested that water mimosa can be a reliable phytobioremediator for polluted water with up to 30 ppm concentrations of arsenic.
Comment 10:
Minor issues:
- There are some misinterpreted statements. See line 172: ‘The developed vacuoles transport water through the cell wall using aquaporin as an advanced transport channel (Toyota et al., 2018). The functions of these channels are similar to those of ion channels.’ These sentences should be reconsidered. The vacuoles itself cannot transport water through the cell walls, the accumulation or release of ions in these vacuoles can help in the regulation of water transport through the plasma membrane with the help of aquaporins (which are very different from other ion channels, in the perspective of the function), not through the cell walls.
Answer:
Many thanks for your constructive comments. We have revise the conclusion and the correct statement is as follow:
The MDA content could increase with heavy metal stress due to the concentration-dependent free radical production. The better the oxidative damage protection is, the more quickly the antioxidative system could be to up-regulate. Peroxidising activity could lead to the MDA Elimination. Increasing the antioxidative enzyme activities such as peroxidising activity lead to the MDA Elimination, subsequently, reduction of H2O2 amounts and membrane damage (Zhan et al. 2007).
Comment 11:
- The units of measure should be standardized also. PPM is used on graphs while mg/L is used in the text.
Answer:
Thank you for the important point. We changed all the mg L-1 to ppm, accordingly.
Comment 12:
- There isn't any mention about the time period of the treatment in the description of table 2.
Answer:
Thank you so much or the important note. We added the term “14 days” and competed the explanation of abbreviation as follow:
Plant growth and physiological parameters were measured on the 14th day of plant treatment. RWC: relative water content, RGR: relative growth rates, and Ti: tolerance index. Different letters indicate significant differences between arsenic concentrations according to Duncan’s multiple range tests (p ≤ 0.05).

Reviewer 3 Report
Dear Authors
The manuscript entitled "Morphological, Physiological and Removal Efficiency Assessment of Water Mimosa (Neptunia Oleracea Lour.) for Phytoremediation of Arsenic Polluted Water" presented the importance of Arsenic phytoremediation which is very important issues. Although the present manuscript needs significant improvement in my opinion. Please find here my suggestion.
1- Language must be improved to make it clearer and long sentences may be avoided, for example line 58-61, "Increases in Malaysia’s population have pushed different industrial, mining and agricultural sectors to try to rapidly develop in order to meet demand; in turn, water sources have become contaminated by various heavy metals, particularly arsenic that are used by these sectors (Jayakumar et al., 2017)". The meaning of the sentence is confusing.
2- Please confirm if reference style match the format of the journal.
3- Introduction is too long, it may be more concise and concrete in terms of phytoremediation of arsenic.
4- Discussion may be improved and conclusion may be drawn clearer and smaller, which may facilitate to convey the key finding.
5- More information regarding collected plants may be incorporated. Study was carried out with naturally grown plants? Phenotypic data of the collected plants? of same age?
6- Only one plant was under treatment with different levels of sodium
heptahydrate arsenate? What about biological/technical replicates?
Thank you
Author Response
Reviewer 3
The manuscript entitled "Morphological, Physiological and Removal Efficiency Assessment of Water Mimosa (Neptunia Oleracea Lour.) for Phytoremediation of Arsenic Polluted Water" presented the importance of Arsenic phytoremediation which is very important issues. Although the present manuscript needs significant improvement in my opinion. Please find here my suggestion.
Comment 1:
- Language must be improved to make it clearer and long sentences may be avoided, for example line 58-61, "Increases in Malaysia’s population have pushed different industrial, mining and agricultural sectors to try to rapidly develop in order to meet demand; in turn, water sources have become contaminated by various heavy metals, particularly arsenic that are used by these sectors (Jayakumar et al., 2017)". The meaning of the sentence is confusing.
Answer:
Many thanks for the reviewer’s concern. The sentence has been edited as followed:
“The rapid increase of human population growth, urbanization, industrial activities, exploration and exploitation of ecosystems have caused heavy metal and metalloid pollutions in in Malaysia's environment”.
On the other hand, we re-sent the draft for editing service again and the current version has been edited by proofreading services.
Comment 2:
- Please confirm if reference style matches the format of the journal.
Answer:
Thank you. All the references have been matched according to the journal format.
Comment 3:
- Introduction is too long, it may be more concise and concrete in terms of phytoremediation of arsenic.
Answer:
Many thanks for the comment. We have implemented the advices as follow, however we are ready if it needs more modification.
- These two sentences are deleted from the introduction of the paragraph “Therefore, evaluation of efficient, simple, low cost and effective systems water remediation is essential (Singh et al., 2018). Remediation strategies should easily overcome environmental degradation problems.”
- These two sentences are merged “For the past two decades, phytoremediation has been developed as a non-invasive and economic alternative to different conventional civil engineering-based strategies for the remediation of water, soil and even residences polluted with heavy metals (Wei et al., 2019). It is an emerging green technology that is efficient in remediating various environmental pollutants (Prabakaran et al., 2019).”
After editing:
- For the past two decades, phytoremediation has been developed as a green, non-invasive and economic alternative to different conventional civil engineering-based strategies for the remediation of water, soil and even residences polluted with heavy metals (Wei et al., 2019; Prabakaran et al., 2019).
As we already have mentioned that phytoremediation is applied for heavy metal treatments, to avoid repetition and redundancy about arsenic as a heavy metal, so, the following sentence is deleted, “The technique has high potential in removing arsenic from aqueous media and has been successfully applied to the metalloid (Dhanwal et al., 2017).”
- These sentences are modified as follow:
Before editing:
Strategies employed under phytoremediation include phytodegradation, phytoaccumulation (or phytoextraction), phytostabilisation, phytofiltration, phytovolatilisation, and Rhizodegradation (Yadav et al., 2018; Prabakaran et al., 2019) (Figure 1). Phytodegradation is a strategy where plants uptake organic xenobiotic substrates and subsequently degrade them using specific enzymes (Arabnezhad et al., 2019). Phytoaccumulation is the process of using roots to uptake contaminants followed by translocating them to aerial tissues where they accumulate for safe disposal (Nasr., 2020). Phytostabilisation is the technology of using plants to reinstate pollutants by decreasing their mobility and bioaccumulation and consequently, prohibiting their locomotion into the nearby food chains and environments environments (Castaldi et al., 2018). Phytofiltration is the process of employing plant biomass to refine contaminants from polluted water systems (Nasr 2020). Phytovolatilisation is a process where contaminants are absorbed by plants which then reform and discharge these toxic substances through less-toxic atmospheric vapours during the transpiration process (Ossai et al., 2020). Rhizodegradation is intensified biodegradation of contaminants associated with rhizosphere microbes’ mediation (Hoang et al., 2020) (Figure 1).
After editing:
Strategies employed under phytoremediation include phytodegradation (which refers employing plant or microorganism to degrade contaminants) (Arabnezhad et al., 2019), phytoaccumulation (which refers employing algae or plant to accumulate contaminants in their areal parts) (Nasr., 2020), phytostabilisation (which refers employing plant to reduce the heavy metal mobility in soil) (Castaldi et al., 2018), phytofiltration (which refers employing plant biomass and their associated rhizospheric microorganisms to refine contaminants) (Nasr 2020), phytovolatilisation (which refers employing plant to absorb contaminants and transpire them into the atmosphere in the volatile shape) (Ossai et al., 2020), and Rhizodegradation (which refers employing plant to degrade contaminants using rhizosphere microbes’ mediation ( Yadav et al., 2018; Prabakaran et al., 2019; Hoang et al., 2020) (Figure 1).
- The following sentences are added “Compared to aboveground parts (leaves and stems), heavy metals could be accumulated in the root of oleracea (Mishra et al., 2008a). The higher accumulation of heavy metals indicated the removal efficiency of this plant through rhizofiltration process (Wahab et al., 2014).”
- The following sentence is also deleted “ oleracea spreads rapidly and is buoyed by spongy, white aerenchyma tissues underneath its stems which float on the water surface (Bhunia and Mondal., 2012).”
We have added the this statement in the draft.
“In addition, the present investigation aims to get insights into the removal efficiency of water mimosa for phytoremediation of arsenic-polluted water.”
Comment 4:
- Discussion may be improved and conclusion may be drawn clearer and smaller, which may facilitate to convey the key finding.
Answer:
Thank you. The discussion has been improved as suggested.
- “Observations on the antioxidant content showed water mimosas’ adequate tolerance under 30 ppm of arsenic treatment for proline”
- The physiological traits analysis showed the toxicity effects of arsenic even at the initial stages of the experiment, which caused damages to the chlorophyll content, photosynthesis rate, stomatal conductance, intercellular co2 concentrations, transpiration rate, and air pressure deficit.
- The inductively coupled plasma optical emission spectrometry (ICP-OES) showed the accumulation of arsenic in water mimosas between the range of 30-60 ppm.
- All in all, the result of this investigation suggested that water mimosa can be a reliable phytobioremediator for polluted water with up to 30 ppm concentrations of arsenic.
Comment 5:
- More information regarding collected plants may be incorporated. Study was carried out with naturally grown plants? Phenotypic data of the collected plants? of same age?
Answer:
Thank you so much for the important point. Initially the plants were collected from a natural pound inside the university. We collected the plants with the same size and weight. We acclimatized them for one month, and after that, new shoots were collected and transferred to a greenhouse under controlled conditions. Therefore they were of the same age and almost the same morphology. Nonetheless, we have to mention that, the number of leaves was not similar in this study, as it was difficult to control the leaves response to touch and other environmental stimuli. However, in the manuscript we just mentioned the collection and acclimatization phase. All in all, we added the “similar size and weight” term in the draft as followed:
Local naturally-grown N. oleracea aquatic plants with similar weight and size were collected from Universiti Putra Malaysia’s pond (2°59'23.8"N, 101°42'46.5"E), and the plant species was confirmed by the Biodiversity Unit (UBD) of Institute of Bioscience, Universiti Putra Malaysia. The collected plants were acclimatized under hydroponic conditions in tanks (12 × 25 × 10 cm = 3 L) containing 0.20x of Hoagland solution with an aeration system.
Additionally, we followed these references to collect the plants from the pound in the first phase of the project: Wahab, A. S. A., Ismail, S. N. S., Praveena, S. M., & Awang, S. (2014). Heavy metals uptake of water mimosa (Neptunia oleracea) and its safety for human consumption. Iranian Journal of Public Health, 43(Supple 3), 103-111.
Comment 6:
6- Only one plant was under treatment with different levels of sodium
heptahydrate arsenate? What about biological/technical replicates?
Answer:
Thank you for the question. As we already mentioned in the “4.12. Statistical Analysis” section “For morphological and physiological studies, a total of 30 tanks were used and distributed in 3 experimental blocks (10 arsenic concentrations × 3 replications).” The biological replicates were 3 and distributed in 3 blocks. However, we completed the section as followed:
For morphological and physiological studies, a Randomized Complete Block Design (RCBD) were used with a total of 30 tanks distributed in 3 experimental blocks (10 arsenic concentrations × 3 replications). To analyse the data, a SAS software version 9.4 was applied. The level of significance was assessed from the analysis of variance (ANOVA). Duncan’s multiple ranges were used to compare the mean values, and interpretations were made accordingly.
Thank you

Round 2
Reviewer 1 Report
Fig. 8 -the caption should be improved (the type of experiment, method or technique used (sample size, type of plant model used, metal concentration, duration of the experiment etc.). It is still uniformative.
Author Response
Dear Reviewer,
Cordial greetings,
Many thanks for your kind attention toward the improvement of our paper. We have completed the mentioned section as follow:
Figure 8. Scanning electron microscopic (SEM) observations were performed on the roots of three independent replications of control and treated (30 ppm sodium heptahydrate arsenate) water mimosa after 14 days of the exposure period. To measure the arsenic contents of root samples, three different spectrums of roots were randomly measured for each image.
Best Wishes,
Narges Atabaki
Reviewer 2 Report
I am happy to see that all my questions were answered.
Moreover, the manuscript itself improved a lot.
Author Response
Dear Reviewer,
Many thanks for your constructive comments, which helped to improve our paper.
Best regards,
Narges Atabaki
Reviewer 3 Report
Dear Authors
I am glad to see the response and manuscript have been significantly improved.
Thank you
Author Response

(The authors gave the same response as above.)
